# Gabapentinoid consumption in 65 countries and regions from 2008 to 2018: a longitudinal trend study

Adrienne Y. L. Chan[1,2,3,4,13], Andrew S. C. Yuen [4,5,13], Daniel H. T. Tsai [6,7], Wallis C. Y. Lau [2,3,4,5], Yogini H. Jani[4,5], Yingfen Hsia[6,8], David P. J. Osborn[9,10], Joseph F. Hayes[9,10], Frank M. C. Besag[4,11], Edward C. C. Lai[7], Li Wei[4,5], Katja Taxis[1], Ian C. K. Wong [2,3,4,5,12,14] ✉ & Kenneth K. C. Man [2,3,4,5,14] ✉

Recent studies raised concerns about the increasing use of gabapentinoids in different countries. With their potential for misuse and addiction, understanding the global consumption of gabapentinoids will offer us a platform to examine the need for any interventional policies. This longitudinal trend study utilised pharmaceutical sales data from 65 countries and regions across the world to evaluate the global trends in gabapentinoid consumption between 2008-2018. The multinational average annual percentage change of gabapentinoid consumption was +17.20%, increased from 4.17 defined daily dose per ten thousand inhabitants per day (DDD/TID) in 2008 to 18.26 DDD/TID in 2018. High-income countries had the highest pooled gabapentinoid consumption rate (39.92 DDD/TID) in 2018, which was more than six times higher than the lower-middle income countries (6.11 DDD/TID). The study shows that despite differences in healthcare system and culture, a consistent increase in gabapentinoid consumption is observed worldwide, with high-income countries remaining the largest consumers.

Gabapentinoids are a class of medications which was first introduced in the United States (US) and the United Kingdom (UK) in 1993[1]. The two major agents, gabapentin and pregabalin, were originally developed as antiseizure agents[2]. The extensive range of clinical actions of gabapentinoids is primarily the result of their inhibitory properties on neuronal voltage-gated calcium channel currents, via impairing the trafficking function of the alpha-2-delta subunits, reducing the signal leading to the release of neurotransmitters[3]. Given their effects on intracellular calcium levels, a considerable proportion of gabapentinoid prescriptions are for the treatment of mental health symptoms and diagnoses, such as insomnia and bipolar disorder, which, however, have limited evidence of efficacy to support their use[2,4].

Licensed indications of gabapentinoids have since expanded to conditions including neuropathic pain, fibromyalgia, postherpetic

[1]Groningen Research Institute of Pharmacy, Unit of Pharmacotherapy, Epidemiology and Economics, University of Groningen, 72 9700 Groningen, The Netherlands. [2]Laboratory of Data Discovery for Health (D24H), Hong Kong Science Park, Hong Kong SAR, China. [3]Centre for Safe Medication Practice and Research, Department of Pharmacology and Pharmacy, Li Ka Shing Faculty of Medicine, The University of Hong Kong, Hong Kong SAR, China. [4]Research Department of Practice and Policy, School of Pharmacy, University College London, London WC1H 9JP, UK. [5]Centre for Medicines Optimisation Research and Education, University College London Hospitals NHS Foundation Trust, London NW1 2PG, UK. [6]Centre for Neonatal and Paediatric Infection, St George's University of London, London SW17 0RE, UK. [7]School of Pharamcy, Institute of Clinical Pharmacy and Pharmaceutical Sciences, College of Medicine, National Cheng Kung University, Tainan, Taiwan. [8]School of Pharmacy, Queen's University Belfast, Belfast BT9 7BL, UK. [9]Division of Psychiatry, University College London, London W1T 7BN, UK. [10]Camden and Islington NHS Foundation Trust, London NW1 0PE, UK. [11]East London Foundation NHS Trust, Bedfordshire MK40 3JT, UK. [12]Aston School of Pharmacy, Aston University, Birmingham B4 7ET, UK. [13]These authors contributed equally: Adrienne Y. L. Chan, Andrew S. C. Yuen. [14]These authors jointly supervised this work: Ian C. K. Wong, Kenneth K. C. Man. ✉e-mail: wongick@hku.hk; kenneth.man@ucl.ac.uk

neuralgia, restless legs syndrome, generalised anxiety disorder, and complications of multiple sclerosis[5–8]. The licensed indications of some of the countries/regions are shown in Supplementary Table 1. In addition to their licensed indications, off-label prescribing of gabapentinoids also accounted for a considerable proportion of their use[9,10]. In the UK, over 50% of gabapentinoid prescriptions are related to off-label indications[9]. In a recent study conducted in the US, around 95% of gabapentin prescriptions in the US were for off-label pain management use and an increasing trend of overlapping use with opioid analgesics has also been observed[11], despite strong evidence suggesting an increase in the risk of all-cause and drug-related hospitalisations from this combination[11,12]. Similar prescribing trends for pain management and off-label indications were also observed in other countries/ regions such as Sweden, Australia and Taiwan, which have different socioeconomic status and healthcare systems when compared to the UK and US[13–15]. The prominent use of gabapentinoids has raised concerns about their potential misuse, which can lead to eventual episodes of hospitalisation or mortality, especially for patients with a history of substance abuse and psychiatric comorbidities[16–18]. A UK study has shown that gabapentinoid-related overdose fatalities have also increased substantially in recent years, and 79% of them also involve the use of opioids[19]. Gabapentin was also reported to be the most misused non-controlled medication in a prison setting in the US[20]. The misuse of gabapentinoids can be explained by not only their euphoric and relaxation effects but also their potential reduction of withdrawal effects of other drugs[18,21,22].

In this study, by analysing validated multinational sales data, we evaluate the worldwide consumption trends of gabapentinoids in 65 countries and regions from 2008–2018. Different from previous studies, which are limited to the national level or individual therapeutic agents[9,11,23–26], the results include 65 countries and regions across the globe and include all gabapenitnoid agents that are available during the studied period. The results indicate an overall increase in gabapentinoid consumption from 2008–2018 across different countries. Pooled regional gabapentinoid consumption rates are highest in North America, followed by Oceania and Northern Europe in 2018. Lower-middle income countries have the largest growth in consumption. Despite differences in healthcare system and culture, a consistent increase in gabapentinoid consumption is observed worldwide, with high-income countries remaining the largest consumers. Given their abuse potential and mixed evidence of off-label uses, international and national regulatory bodies may review current guidelines towards the use gabapentinoids.

## Results

Among the 65 countries/regions, representing approximately 70% of the global population, there was an overall increase in gabapentinoid consumption from 2008–2018 (Table 1 and Figs. 1, 2, 3). The average annual percentage change of gabapentinoid consumption was +17.20% (95%CI, +15.52% to +18.91%), from 4.17 DDD/TID (95%CI, 2.99 to 5.81) in 2008 to 18.26 DDD/TID (95%CI, 13.54 to 24.63) in 2018 (Table 1). The characteristics of included countries and the availability of different gabapentinoids sold were presented in Supplementary Table 3. Gabapentin and pregabalin were available in all studied countries whereas gabapentin enacarbil, a prodrug of gabapentin, was only sold in the US, Puerto Rico and Japan.

Across regions, upward trends of gabapentinoid consumption from 2008-2018 were consistently observed (Table 1 and Figs. 2, 3). Average annual increase in consumption was the highest in Northern Africa (+35.91; 95%CI, +26.17% to +46.41%), followed by Eastern Asia (+28.51%; 95%CI, +18.86% to +38.94%), Eastern Europe (+23.77%; 95%CI, +17.06% to +30.86%), Central Asia, (+20.45%; 95%CI, −0.53% to +45.85%), Oceania (+19.89%; 95%CI, +13.70% to +26.43%), Western Asia (+17.57%; 95%CI, +10.53% to +25.06%), Southern Asia (+15.56%; 95%CI, +12.19% to +19.03%), Southern Europe (+14.91%; 95%CI, +10.92% to

+19.05%), Northern Europe (+14.78%; 95%CI, +12.34% to +17.27%), South-eastern Asia (+14.70%; 95%CI, +9.50% to +20.04%), Central and Southern America and the Caribbean (+12.92%; 95%CI, +9.55% to +16.39%), Southern Africa (+12.42%; 95%CI, +6.35% to +18.85%), Northern America (+9.04%; 95%CI, +6.82% to +11.32%), and Western Europe (+8.15%; 95%CI, +6.64% to +9.68%). In 2018, pooled gabapentinoid consumption rates were the highest in Northern America (124.62 DDD/TID; 95%CI, 95.77 to 162.16), followed by Oceania (68.88 DDD/TID; 95%CI, 37.14 to 127.72), and Northern Europe (54.66 DDD/TID; 95%CI, 38.59 to 77.43). The gabapentinoid consumption was lowest in Central Asia (1.05 DDD/TID; 95%CI, 1.04 to 1.05; Table 1 and Fig. 3).

At country level, except for Venezuela, rising trends in gabapentinoid consumption were reported in all countries from 2008-2018 (Table 1 and Fig. 4). In 2018, the top three countries/ territories that had the highest consumption of gabapentinoids were: Puerto Rico (151.23 DDD/TID; 95%CI, 151.16 to 151.30), the US (142.54 DDD/TID; 95%CI, 142.53 to 142.55), and the UK (138.88 DDD/TID; 95%CI, 138.86 to 138.89) (Table 1). Results from the sensitivity analyses were similar to the main analysis (Supplementary Tables 4, 5). The other sensitivity analysis that removed products with imputed strength has a pooled multinational consumption levels of 4.11 DDD/TID (95%CI, 2.99 to 5.66) in 2008 and 17.84 DDD/TID (95%CI, 13.39 to 23.77) in 2018, which are comparable to our main analysis.

When stratified by income levels, the annual average increase of gabapentinoid consumption was the highest in lower-middle income countries ($n = 6$; +23.28%; 95%CI, +18.55% to +28.21%), followed by upper-middle countries ($n = 21$; +21.98%; 95%CI, +17.99% to +26.10%), and high-income countries ($n = 38$; +13.84%; 95%CI, +12.18% to +15.53%) (Table 2). However, high-income countries still have considerably higher consumption than lower-middle and upper-middle countries throughout the study period. In 2018, the pooled consumption rates of gabapentinoids were 39.92 DDD/TID (95%CI, 32.35 to 49.26) in high-income countries, 6.06 DDD/TID (95%CI, 3.15 to 11.66) in upper-middle-income countries and 6.11 DDD/TID (95%CI, 2.12 to 17.61) in lower-middle-income countries (Table 2).

Among the three gabapentinoids, gabapentin enacarbil had the greatest multinational increase in DDD/TID during the study period, with an average annual percentage change of +35.86% (95% CI, +13.47% to +62.66%), followed by pregabalin (+23.14%; 95%CI, +20.07% to +26.28%) and gabapentin (+8.70%; 95%CI, +6.86% to +10.56%). The average annual changes for different gabapentinoids from 2008–2018 multinationally, by region, and by country are available in Supplementary Tables 6–8. At regional level, Northern America had the highest consumption of gabapentin (75.09 DDD/TID; 95%CI, 31.55 to 178.71), Eastern Asia had the highest consumption of gabapentin enacarbil (0.63 DDD/TID; 95%CI, 0.63 to 0.64), Western Europe had the highest consumption of pregabalin (39.87 DDD/TID, 36.21 to 43.90). At country/territory-level, Puerto Rico had the highest consumption of gabapentin (139.25 DDD/TID; 95%CI 139.18 to 139.32), Japan had the highest consumption of gabapentin enacarbil (0.63 DDD/TID; 95%CI, 0.63 to 0.64), and Australia had the highest consumption of pregabalin (87.38 DDD/TID; 95%CI, 87.36 to 87.40; Fig. 2).

## Discussion

Our study reported the consumption of gabapentinoids in 65 countries and regions, classified by country income levels and geographical regions. The results showed a substantial increase in multinational gabapentinoid consumption over the span of 11 years, with more than four-fold increase in DDD/TID from 2008-2018 and an average increase of 17.20% per annum. The rise in consumption of gabapentinoids across the globe was consistent with previous national studies in the US, UK and Canada[9,11,23–27].

Even though the first gabapentinoid had been marketed since 1993[26], the momentum of consumption growth was maintained

**Table 1 | Multinational, regional and national levels of gabapentinoid consumption in 2008 and 2018 and average annual percentage change**

| | DDD/TID in 2008 (95%CI)[a] | DDD/TID in 2018 (95%CI)[a] | Average annual percentage change (%, 95%CI)[b] |
|---|---|---|---|
| **Multinational** | 4.17 (2.99, 5.81) | 18.26 (13.54, 24.63) | 17.20 (15.52, 18.91) |
| **America (North)** | 52.44 (32.79, 83.86) | 124.62 (95.77, 162.16) | 9.04 (6.82, 11.32) |
| Canada | 41.27 (41.25, 41.28) | 108.95 (108.94, 108.97) | 10.59 (9.70, 11.49) |
| United States | 66.63 (66.63, 66.64) | 142.54 (142.53, 142.55) | 7.89 (6.82, 8.97) |
| **America (Central and Southern) and the Caribbean** | 2.33 (0.90, 6.02) | 7.86 (3.51, 17.58) | 12.92 (9.55, 16.39) |
| Argentina | 2.11 (2.11, 2.11) | 13.90 (13.90, 13.91) | 20.53 (14.57, 26.80) |
| Brazil | 0.52 (0.52, 0.52) | 6.32 (6.32, 6.32) | 26.82 (22.90, 30.86) |
| Chile | 2.11 (2.11, 2.12) | 9.28 (9.27, 9.29) | 17.95 (16.07, 19.85) |
| Colombia | 0.88 (0.87, 0.88) | 1.96 (1.95, 1.96) | 10.48 (7.45, 13.61) |
| Ecuador | 1.95 (1.94, 1.95) | 5.48 (5.47, 5.48) | 10.27 (8.31, 12.28) |
| Mexico | 2.79 (2.79, 2.79) | 4.13 (4.12, 4.13) | 3.84 (1.93, 5.78) |
| Peru | 0.50 (0.50, 0.50) | 2.56 (2.56, 2.56) | 19.06 (16.71, 21.47) |
| Puerto Rico | 50.09 (50.05, 50.13) | 151.23 (151.16, 151.30) | 12.33 (10.51, 14.17) |
| Uruguay | 2.54 (2.53, 2.55) | 15.40 (15.38, 15.42) | 19.16 (12.91, 25.75) |
| Venezuela | 6.79 (6.79, 6.80) | 4.18 (4.18, 4.19) | -3.70 (–11.98, 5.36) |
| **Europe (West)** | 23.77 (19.12, 29.55) | 50.85 (46.14, 56.05) | 8.15 (6.64, 9.68) |
| Austria | 21.98 (21.96, 21.99) | 60.50 (60.48, 60.53) | 10.41 (8.93, 11.91) |
| Belgium | 13.01 (13.00, 13.02) | 56.26 (56.24, 56.29) | 14.62 (12.07, 17.23) |
| France | 35.47 (35.46, 35.47) | 60.18 (60.17, 60.19) | 5.39 (4.72, 6.07) |
| Germany | 27.85 (27.84, 27.85) | 56.64 (56.63, 56.64) | 6.93 (5.83, 8.04) |
| Luxembourg | 32.51 (32.43, 32.60) | 44.24 (44.15, 44.33) | 3.11 (2.37, 3.87) |
| Netherlands | 0.00 (0.00, 0.00) | 40.54 (40.52, 40.55) | 6.78 (5.39, 8.18) |
| Switzerland | 19.62 (19.61, 19.64) | 42.26 (42.24, 42.28) | 8.11 (7.03, 9.20) |
| **Europe (North)** | 13.77 (11.03, 17.21) | 54.66 (38.59, 77.43) | 14.78 (12.34, 17.27) |
| Estonia | 2.02 (2.01, 2.03) | 30.00 (29.95, 30.05) | 29.97 (27.75, 32.23) |
| Finland | 43.56 (43.53, 43.59) | 82.83 (82.79, 82.87) | 5.73 (4.61, 6.87) |
| Ireland | 28.73 (28.70, 28.75) | 85.54 (85.50, 85.58) | 13.75 (10.33, 17.27) |
| Latvia | 4.16 (4.15, 4.18) | 26.39 (26.36, 26.43) | 21.02 (19.53, 22.53) |
| Lithuania | 3.47 (3.46, 3.48) | 20.12 (20.09, 20.15) | 19.49 (16.96, 22.07) |
| Norway | 34.30 (34.27, 34.33) | 66.32 (66.28, 66.36) | 6.82 (6.07, 7.57) |
| Sweden | 33.98 (33.96, 34.00) | 76.67 (76.64, 76.70) | 7.99 (7.35, 8.62) |
| United Kingdom | 30.46 (30.45, 30.46) | 138.88 (138.86, 138.89) | 16.99 (14.83, 19.19) |
| **Europe (South)** | 16.40 (10.55, 25.48) | 27.10 (18.20, 40.34) | 14.91 (10.92, 19.05) |
| Bosnia And Herzegovina | 0.00 (0.00, 0.00) | 5.33 (5.32, 5.35) | 35.56 (27.42, 44.22) |
| Croatia | 3.16 (3.15, 3.17) | 13.82 (13.80, 13.83) | 15.53 (12.92, 18.21) |
| Greece | 22.48 (22.47, 22.50) | 46.34 (46.32, 46.36) | 6.44 (5.24, 7.65) |
| Italy | 14.92 (14.91, 14.92) | 28.82 (28.81, 28.82) | 6.64 (5.64, 7.65) |
| Portugal | 31.22 (31.21, 31.24) | 51.52 (51.50, 51.55) | 5.10 (3.82, 6.40) |
| Serbia | 0.00 (0.00, 0.00) | 19.56 (19.54, 19.57) | 57.97 (48.8, 67.70) |
| Slovenia | 15.31 (15.28, 15.34) | 40.00 (39.95, 40.04) | 10.06 (9.17, 10.95) |
| Spain | 38.35 (38.34, 38.36) | 73.28 (73.27, 73.29) | 6.17 (4.91, 7.44) |
| **Europe (East)** | 1.76 (0.70, 4.42) | 14.88 (7.27, 30.44) | 23.77 (17.06, 30.86) |
| Belarus | 0.06 (0.06, 0.06) | 1.87 (1.86, 1.87) | 35.74 (30.09, 41.63) |
| Bulgaria | 1.05 (1.05, 1.06) | 13.77 (13.76, 13.79) | 24.48 (18.37, 30.90) |
| Czech Republic | 11.42 (11.41, 11.43) | 60.55 (60.52, 60.57) | 17.84 (16.42, 19.27) |
| Hungary | 7.21 (7.20, 7.22) | 21.51 (21.49, 21.52) | 11.43 (9.05, 13.87) |
| Poland | 1.51 (1.51, 1.51) | 16.80 (16.79, 16.80) | 26.34 (19.72, 33.32) |
| Romania | 3.11 (3.10, 3.11) | 16.37 (16.36, 16.37) | 15.43 (12.31, 18.64) |
| Russia | 0.29 (0.29, 0.29) | 5.05 (5.05, 5.05) | 32.16 (11.78, 56.24) |
| Slovakia | 14.11 (14.10, 14.13) | 51.63 (51.59, 51.66) | 12.87 (9.87, 15.94) |
| **Oceania** | 11.27 (8.69, 14.61) | 68.88 (37.14, 127.72) | 19.89 (13.70, 26.43) |
| Australia | 12.86 (12.86, 12.87) | 94.38 (94.36, 94.40) | 26.67 (21.30, 32.27) |
| New Zealand | 9.87 (9.85, 9.88) | 50.26 (50.23, 50.29) | 16.61 (15.77, 17.45) |

**Table 1 (continued) | Multinational, regional and national levels of gabapentinoid consumption in 2008 and 2018 and average annual percentage change**

| | DDD/TID in 2008 (95%CI)[a] | DDD/TID in 2018 (95%CI)[a] | Average annual percentage change (%, 95%CI)[b] |
|---|---|---|---|
| **Asia (East)** | 0.63 (0.10, 4.18) | 7.74 (1.31, 45.89) | 28.51 (18.86, 38.94) |
| China | 0.01 (0.01, 0.01) | 0.44 (0.44, 0.44) | 45.23 (36.07, 55.00) |
| Japan | 1.02 (1.01, 1.02) | 41.32 (41.32, 41.33) | 45.79 (28.56, 65.32) |
| South Korea | 9.48 (9.47, 9.48) | 30.99 (30.99, 31.00) | 12.12 (11.07, 13.17) |
| Taiwan | 1.75 (1.75, 1.75) | 6.41 (6.41, 6.42) | 15.08 (13.02, 17.18) |
| **Asia (Central)** | 0.00 (0.00, 0.00) | 1.05 (1.04, 1.05) | 20.45 (−0.53, 45.85) |
| Kazakhstan | 0.00 (0.00, 0.00) | 1.05 (1.04, 1.05) | 18.37 (11.16, 26.05) |
| **Asia (West)** | 3.12 (1.38, 7.06) | 15.73 (6.40, 38.69) | 17.57 (10.53, 25.06) |
| Jordan | 1.27 (1.26, 1.27) | 6.51 (6.50, 6.52) | 24.09 (17.99, 30.51) |
| Kuwait | 1.18 (1.17, 1.18) | 32.25 (32.22, 32.28) | 40.64 (34.36, 47.22) |
| Lebanon | 4.02 (4.01, 4.03) | 15.69 (15.67, 15.70) | 13.02 (10.93, 15.15) |
| Saudi Arabia | 3.62 (3.61, 3.62) | 11.36 (11.35, 11.36) | 16.57 (11.01, 22.40) |
| Türkiye | 13.87 (13.86, 13.87) | 60.98 (60.97, 60.99) | 15.81 (13.63, 18.04) |
| United Arab Emirates | 3.10 (3.09, 3.11) | 6.64 (6.63, 6.65) | 6.55 (−0.91, 14.57) |
| **Asia (South-east)** | 0.53 (0.53, 0.53) | 3.84 (0.67, 22.12) | 14.70 (9.50, 20.00) |
| Philippines | 0.53 (0.53, 0.53) | 1.57 (1.57, 1.58) | 10.40 (9.50, 11.40) |
| Thailand | 0.00 (0.00, 0.00) | 9.39 (9.38, 9.39) | 16.50 (12.00, 21.20) |
| **Asia (South)** | 1.06 (0.90, 1.25) | 4.53 (3.54, 5.78) | 15.56 (12.19, 19.03) |
| India | 1.15 (1.15, 1.15) | 5.13 (5.13, 5.13) | 14.83 (12.63, 17.08) |
| Pakistan | 0.97 (0.97, 0.97) | 3.99 (3.99, 3.99) | 13.21 (11.07, 15.39) |
| **Africa (North)** | 0.68 (0.41, 1.14) | 14.77 (8.49, 25.70) | 35.91 (26.17, 46.41) |
| Algeria | 0.71 (0.71, 0.71) | 29.44 (29.43, 29.44) | 36.74 (27.43, 46.72) |
| Egypt | 1.06 (1.06, 1.07) | 39.54 (39.54, 39.55) | 39.14 (35.14, 43.27) |
| Morocco | 0.24 (0.23, 0.24) | 2.01 (2.01, 2.01) | 20.52 (17.94, 23.16) |
| Tunisia | 1.22 (1.22, 1.23) | 20.33 (20.31, 20.34) | 24.68 (18.90, 30.74) |
| **Africa (South)** | 1.16 (1.16, 1.16) | 3.74 (3.74, 3.74) | 12.42 (6.35, 18.85) |
| South Africa | 1.16 (1.16, 1.16) | 3.74 (3.74, 3.74) | 10.88 (8.73, 13.07) |

*CI* confidence interval, *DDD/TID* defined daily dose per 10,000 inhabitants per day.

[a]Worldwide and regional estimates with 95% CI were calculated by pooling the estimates using meta-analysis (random-effects model).

[b]The average annual percentage change is calculated using a linear regression model, with log-transformed consumption in DDD/TID as the dependent variable and year as the independent variable. The average annual change was expressed as average annual percentage change, calculated by [exp(the coefficient of the year variable) – 1] ×100%. The multinational and regional trend changes were estimated using linear mixed models, controlling for within-country correlations and assuming the correlations between years were autocorrelated.

throughout the study period, despite the differences in income level, geographical location, healthcare system and culture. Venezuela was unique amongst all the countries studied in that there was not an increase in consumption. This different result may be attributed to its economic downturn and reduced health expenditure which began in the early 2010s, causing nationwide shortages of medicinal supply[28]. The consumption of gabapentinoids from upper-middle and lower-middle countries increased at a faster rate than in high-income countries. Nevertheless, by the end of 2018, the pooled consumption from high-income countries was still more than six-fold greater than that of upper-middle and lower-middle countries, which demonstrates disparities in gabapentinoid consumption across the globe. The observed difference is consistent with the greater rate of pain treatment among high-income countries reported in a previous study[29].

One of the major driving factors of the overall increase in gabapentinoid consumption is likely to be their wide range of on and off-label indications[2,4,30]. Given that off-label prescriptions of gabapentinoids have been reported to have accounted for more than half of total prescriptions in many countries, such as the UK and the US[9,31], the increasing trends observed in our study raises the concern of off-label use of gabapentinoids which has also increased rapidly in recent years. Off-label prescribing may be attributed to pharmaceutical companies' marketing campaigns, which have been reported to have made

misleading claims in the past[32]. We cautiously recommend the need for revisiting the appropriateness of the prescriptions issued by clinicians since off-label prescribing of gabapentinoids may often be based on clinical experience with mixed or limited evidence[4,33].

Another potential reason for the increase in gabapentinoid consumption is the increasing concerns over misuse of opioids and benzodiazepines[34–37]. Gabapentinoids have often been perceived as safer alternatives[1,34–37]. In some of the studied countries, the less stringent requirements for prescription compared to opioids and benzodiazepines, might have increased the ease of access to gabapentinoids for prescribers and patients. Clinicians, now more aware of the dependence and overdose issues with opioids and benzodiazepines, might have turned to prescribe gabapentinoids for the treatment of neuropathic pain, generalised anxiety, and insomnia[4,38]. However, a profound level of co-prescribing with opioids and benzodiazepines has been observed, increasing the risk of life-threatening central nervous system and respiratory depression[9]. Increasing concerns have been expressed with regard to abuse and dependence risk of gabapentinoids with or without other medications[16–18,39]. Patients who have a history of opioid, benzodiazepine, or alcohol misuse have been reported as being vulnerable to misuse of gabapentinoids[12,16,24,40]. The rise of gabapentinoid consumption may represent another medication misuse crisis[41].

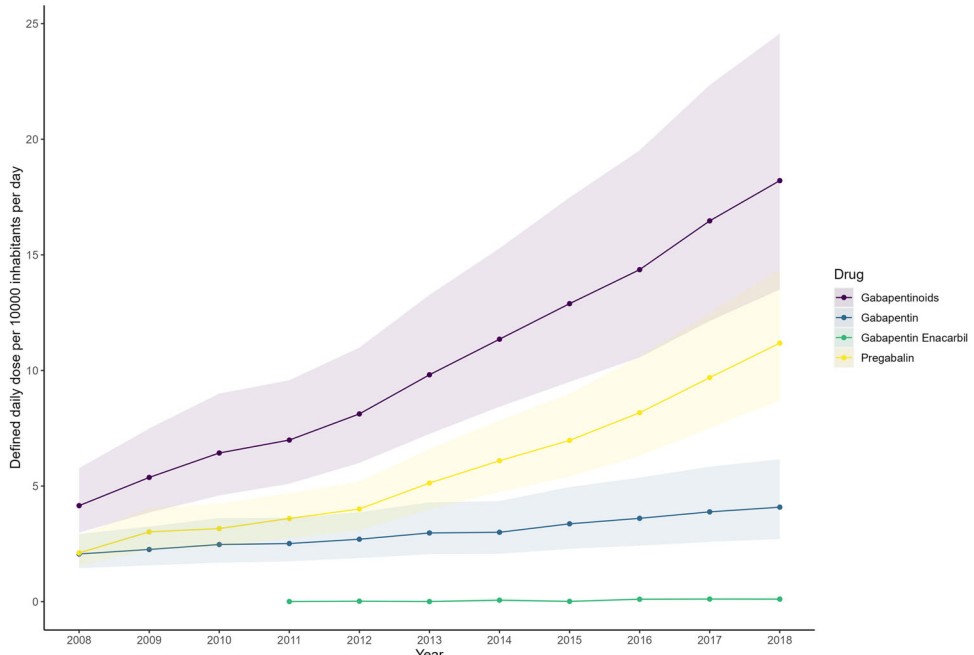

**Fig. 1 | Multinational gabapentinoid consumption from 2008 to 2018.** Multinational consumption levels are presented as pooled defined daily dose per 10,000 inhabitants per day with the shaded areas indicating the 95% confidence bands. The multinational consumption levels were computed by pooling the estimates from individual countries using a random-effects model. The multinational consumption levels of different gabapentinoid agents are presented with 95% confidence interval and different colours. Source data are provided as a Source Data file.

When comparing the change in consumption of the three gabapentinoids studied, all included regions except for Northern America, recorded a stronger growth in annual change in the consumption of pregabalin and gabapentin enacarbil than gabapentin. Despite its valid patent protection till 2015 in Europe and its higher cost of treatment, the general DDD/TID of pregabalin across all regions was recorded to be higher than that of gabapentin. Contributing factors that might account for the increasing consumption of pregabalin since the start of the study period include its twice-daily dosage regimen and more rapid absorption in contrast to gabapentin[38,42]. Pregabalin's preferable cost effectiveness in treating neuropathic pain can also be a driving factor for its higher consumption[43,44]. However, its rapid absorption and steeper dose-response relationship would also lead to a greater overdose potential, of which clinicians should be aware[16].

The weaker growth of pregabalin in the US may be explained by its federal schedule V controlled substance classification since 2005[45], which is not the case for gabapentin. This created legal barriers to prescribing pregabalin in the US which drove the lower growth in comparison to the neighbouring country of Canada. This barrier in accessing pregabalin had led to comparatively high consumption of gabapentin, which was the tenth most commonly prescribed medication in the US in 2017[27]. As shown in Table 1 and Fig. 2, Puerto Rico and the US had the highest consumption rate of gabapentinoids in 2018. They also share a similar trend in gabapentinoid consumption across the years, with gabapentin taking up the majority of the increase in consumption. This phenomenon may be explained by the fact that Puerto Rico is a territory of the US. Many of Puerto Rico's healthcare professionals are trained in the US and may share similar clinical beliefs as their US counterparts. A similar difference was observed between Australia and New Zealand. Unlike Australia, where the increase in gabapentinoid consumption was mainly driven by pregabalin, the increase in New Zealand was mainly due to the rise of gabapentin prescribing. The opposite trend of growth may have been caused by the decision of the Pharmaceutical Management Agency (Pharmac), the pharmaceutical management agency in New Zealand[46], not to fund the use of pregabalin until December 2017[47]. With the recent change of legal classifications for gabapentinoids in the UK to Schedule 3 controlled drugs in 2019[48], further assessments of the effect of this legal change are warranted. Future comparison between the consumption of gabapentinoids in the UK and other neighbouring European countries will also provide us with an insight into the impact on gabapentinoid consumption of their different regulatory approaches.

International pharmaceutical sales data from Multinational Integrated Data Analysis System (MIDAS) offers a platform for global comparison of gabapentinoid consumption among different healthcare systems. Previous studies mainly focus on the number of prescriptions and concomitant use with opioids in individual countries[9,11,23,24]. This study will be the first study to report on the global trends in the consumption of gabapentinoids. The use of DDD/TID can also offer us insight into the absolute consumption of gabapentinoids. However, our study has limitations. First, the MIDAS database is a sales database, information on patient's age, gender, duration of prescription, medical diagnosis and concomitant medications is unavailable. We were unable to investigate the major clinical indications that are contributing to the rise in consumption of gabapentinoids. Individual patient data will be needed to study the appropriateness of gabapentinoid prescriptions or whether there was an increasing trend of misuse correlated to the growth in consumption. Second, the database provides sales figures related only to legitimate means of distribution. Illicit sales of gabapentinoids are not captured and consequently the data presented in the study may not reflect the pattern of overall consumption in countries covered. Third, although 70% of the world population were included in our study, the findings do not necessarily apply to the countries that are not included in the dataset. Fourth, gabapentinoid consumption could be underestimated in countries without 100% market coverage despite adjustments made to project the total consumption, especially in countries that did not have hospital coverage. However, total pharmaceutical market coverage in most countries was greater than 80%. This may affect the estimation of consumption levels but unlikely to influence the estimation of trends

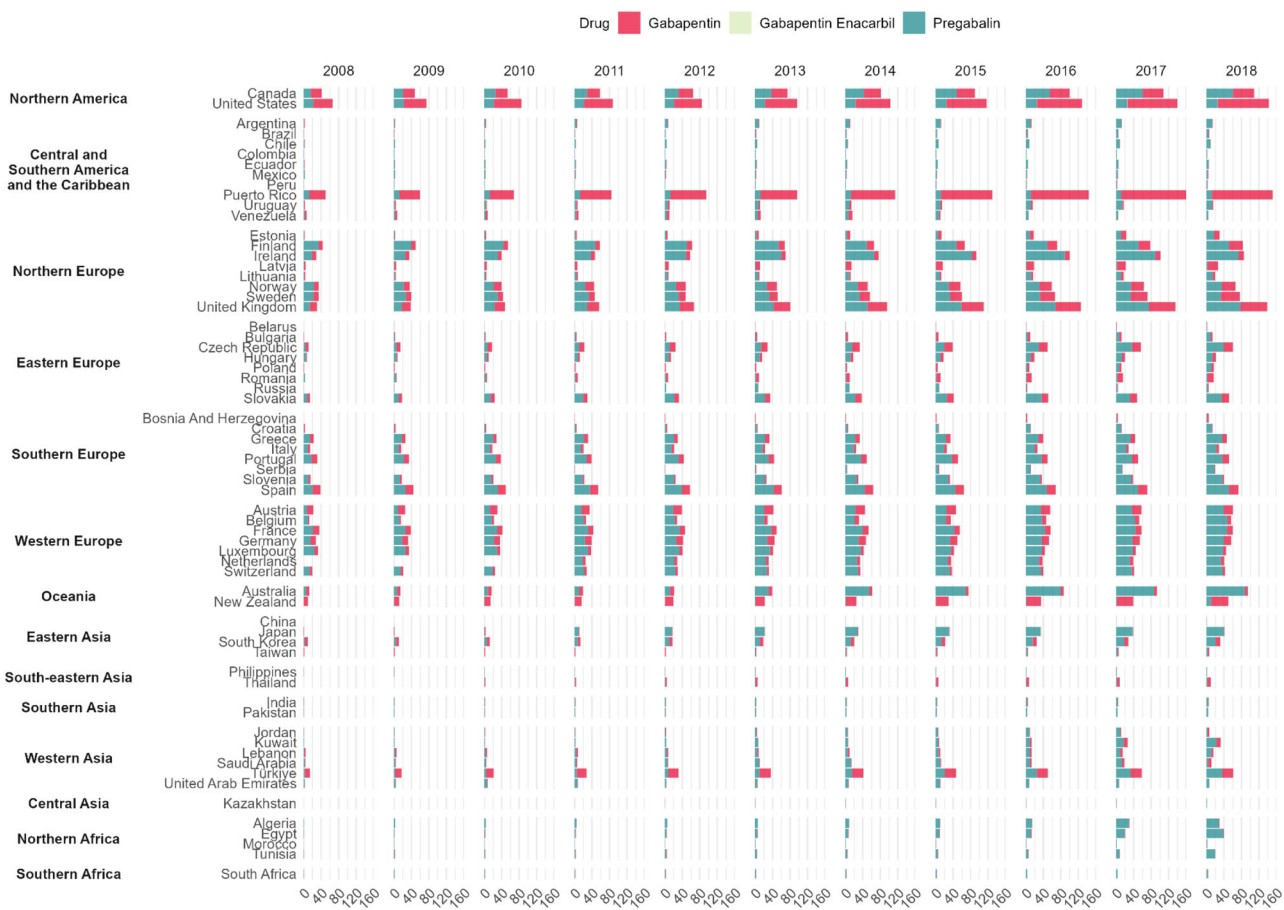

**Fig. 2 | Consumption levels of different gabapentinoids by countries from 2008 to 2018.** The consumption levels of gabapentin, pregabalin and gabapentin ena-carbil of the studied countries from 2008 to 2018 are presented. Each agent is represented by a different colour. The studied countries are grouped according to their geographical locations. Source data are provided as a Source Data file.

since it is a relative measure. Our sensitivity analysis using only retail data showed that the lack of hospital coverage only affected the estimates for individual countries and did not significantly affect the multinational or regional consumption levels. Last but not least, as this is a descriptive study with potential for unexplored variables, we cannot conclude a causal link between any factors and our observed gabapentinoid consumption trends. Other potential factors including but not limited to on and off-label indications, presence of generics, cost of drug, healthcare system, reimbursement status and relevant guidelines, (Supplementary Tables 1, 3, 9–11) may affect the drug utilisation pattern. To accurately assess the impact of these factors, a dedicated study utilising a single, comprehensive database platform is necessary. It is noteworthy that MIDAS, which focuses on national medication consumption data, does not provide sufficient information to thoroughly investigate the effects of these additional factors.

With pregabalin losing its patent in 2015 and less expensive generics beginning to be available in different countries, the overall consumption of gabapentinoids could increase further. A number of regulatory bodies have started to alter their approach towards the control of gabapentinoids, including a change in legal scheduling, change of treatment guidelines and labelling requirements[45,48–50]. Results from the current study, particularly in lower-middle-income and upper-middle-income countries, should be used to support plans for future national, regional, and global public health policies. In view of the increasing concerns over their dependence and misuse potential, further studies are also needed to monitor the safety and appropriateness of gabapentinoid use and to investigate the potential predictors of the increase in gabapentinoid use.

The consumption of gabapentinoids has increased significantly over the span of 11 years, from 2008-2018, inclusive. This increasing trend has been consistent among countries from all income levels. High-income countries remained to be the largest consumers of gabapentinoids, whereas upper-middle-income and lower-middle-income countries showed greater growth in consumption. Against the background of this evidence of increasing use, considering both the abuse potential and the mixed evidence for off-label use, further studies are warranted to investigate the implications behind the increase in consumption and if there is a case for international and national regulatory bodies to review existing treatment guidelines and public health policy relating to gabapentinoids.

## Methods
### Data sources
This study utilised multinational sales data from the IQVIA-MIDAS database. IQVIA is a company specialising in healthcare analytic data. MIDAS captures multinational data on sales volumes of pharmaceutical products from various distribution channels including manufacturers, wholesalers, hospitals, and retail pharmacies, and applies international standardisation to allow comparisons of national sales volumes. The average national coverage of MIDAS data is 88%[51–53]. For countries where the MIDAS database did not have 100% sector coverage, adjustments were made by IQVIA to estimate the total sales

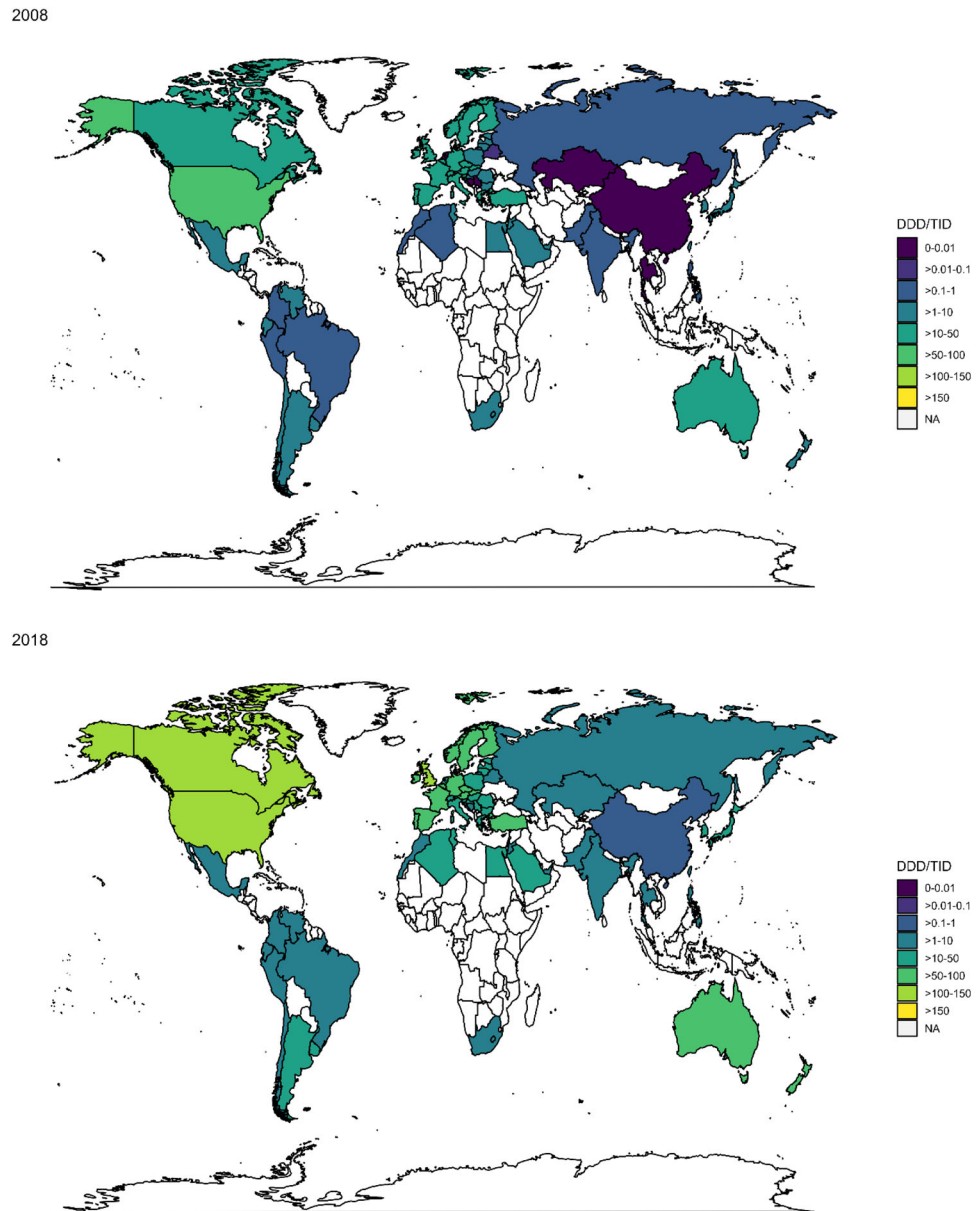

**Fig. 3 | Gabapentinoid consumption in DDD/TID in 2008 and 2018.** Gabapentinoid consumption of the studied countries in 2008 and 2018 are highlighted. Their consumption levels are represented by different colours. Source data are provided as a Source Data file. DDD/TID−defined daily dose per 10,000 inhabitants per day; NA data not available.

volume based on knowledge of the market share of participating wholesalers and retail or hospital pharmacies[54,55]. The MIDAS database has been validated against external data sources[56] and used as a proxy to evaluate multinational consumption of different classes of medications[52,55,57–60]. With the similar approach in previous studies, we utilised the sales data to investigate the consumption of the medication by patients in each country[51,55]. The MIDAS database does not contain patient-level data; thus, no information on patient demographics was available and institutional review board approval was not required.

**Data inclusion**

Data on the sales of gabapentinoids between 2008 and 2018 were extracted from 65 countries and regions in the IQVIA-MIDAS database. The gabapentinoids in this study included gabapentin, gabapentin enacarbil (a prodrug of gabapentin), and pregabalin. Since mirogabalin was only marketed in January 2019[61], it was not included in our study.

The included countries were divided into the following areas: Northern America, Central and Southern America and the Caribbean, Northern Europe, Eastern Europe, Southern Europe, Western Europe, Oceania, Eastern Asia, Central Asia, South-eastern Asia, Southern Asia, Western Asia, Northern Africa, and Southern Africa, based on their geographical regions according to United Nations' (UN) Standard Country or Area Codes for Statistical Use[62]. The mid-year population estimates of each country was obtained from the UN Population Division in 2019[63].

**Statistical analysis**

The main outcome metric was the rate of gabapentinoid consumption, expressed as the defined daily dose (DDD) per ten-thousand inhabitants per day (DDD/TID). DDD is the assumed average maintenance dose per day for a drug used for its main indication in adults and was only available for single-molecule products[64]. As such, DDD count for combination products was converted from a standard unit (defined as a single tablet, capsule, or ampoule/vial or 5 mL oral

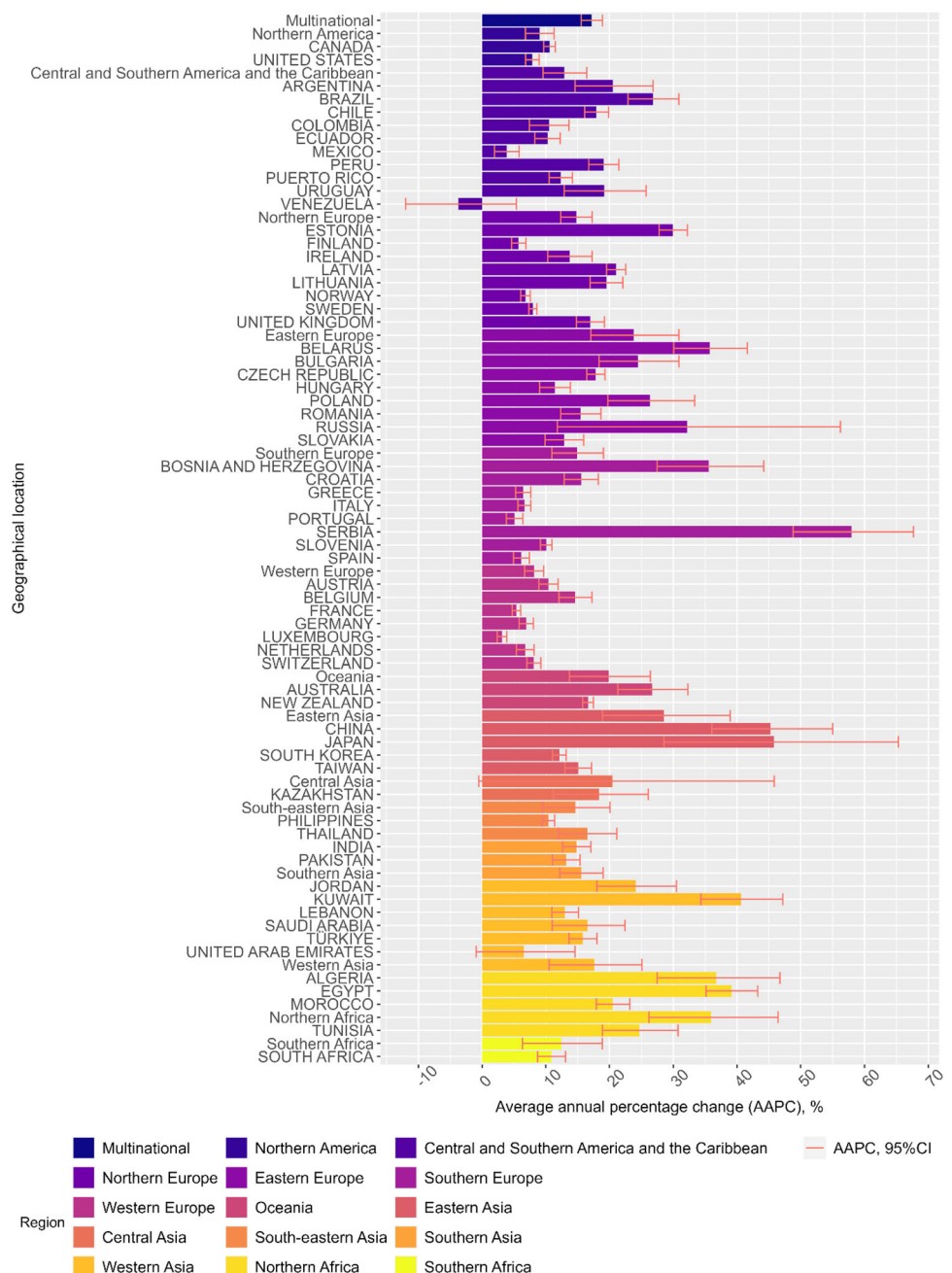

**Fig. 4 | Average annual percentage change of gabapentinoid consumption.**
Different countries/regions' average annual percentage change of gabapentinoid consumption are represented ± 95% confidence interval (error bar). Each country/region is represented with a different colour. The average annual percentage change is calculated using a linear regression model, with log-transformed consumption in DDD/TID as the dependent variable and year as the independent variable. The average annual change was expressed as average annual percentage change, calculated by [exp(the coefficient of the year variable) − 1] × 100%. The multinational and regional trend changes were estimated using linear mixed models, controlling for within-country correlations and assuming the correlations between years were autocorrelated. Source data are provided as a Source Data file.

solution/ suspension), formulation, with their respective drug ingredients mapped to the Anatomical Therapeutic Chemical (ATC)/DDD Index developed by the World Health Organisation (WHO) Collaborating Centre for Drug Statistics Methodology (Supplementary Table 2)[65]. Where the strength or formulation of the product was missing, they were imputed based on the respective information of the most-sold product of the same drug[53].

At the national level, consumption rates in DDD/TID were calculated with a 95% confidence interval (CI) estimated by the Poisson method[66]. The multinational and regional consumption levels were computed by pooling the estimates from individual countries using a random-effects model[67]. The time trends of gabapentinoid consumption were evaluated at multinational, regional, and national levels across the study period. At the national level, the average annual percentage change in DDD/TID with 95% CI was estimated using a linear regression model, with log-transformed consumption in DDD/TID as the dependent variable and year as the independent variable. The multinational and regional trend changes were estimated using linear mixed models, controlling for within-country correlations and accounting for first-order autocorrelation between years. We further

**Table 2 | Annual pooled gabapentinoid consumption and average annual percentage change from 2008 to 2018 by country income level**

| Income level | High (n = 38) | Upper-middle (n = 21) | Lower-middle (n = 6) |
|---|---|---|---|
| Average annual percentage change[a] (%, 95% CI) | 13.84 (12.18, 15.53) | 21.98 (17.99, 26.10) | 23.28 (18.55, 28.21) |
| Year | Pooled consumption (defined daily dose per 1000 inhabitants per day) | | |
| 2008 | 10.79 (8.45, 13.78) | 0.96 (0.48, 1.93) | 0.75 (0.59, 0.96) |
| 2009 | 12.79 (10.06, 16.26) | 1.45 (0.72, 2.93) | 1.08 (0.85, 1.38) |
| 2010 | 15.33 (11.99, 19.60) | 1.85 (0.93, 3.68) | 1.32 (1.00, 1.73) |
| 2011 | 18.30 (14.53, 23.06) | 1.89 (0.99, 3.60) | 1.60 (1.19, 2.14) |
| 2012 | 20.38 (16.58, 25.05) | 2.34 (1.22, 4.47) | 1.93 (1.37, 2.73) |
| 2013 | 23.94 (19.48, 29.42) | 3.01 (1.65, 5.49) | 2.25 (1.53, 3.31) |
| 2014 | 27.14 (22.23, 33.15) | 3.64 (2.01, 6.59) | 2.52 (1.63, 3.88) |
| 2015 | 30.03 (24.37, 37.01) | 4.31 (2.31, 8.03) | 2.92 (1.79, 4.76) |
| 2016 | 34.12 (27.60, 42.17) | 4.58 (2.32, 9.02) | 3.38 (1.86, 6.13) |
| 2017 | 37.82 (30.51, 46.88) | 5.47 (2.80, 10.69) | 4.15 (1.93, 8.94) |
| 2018 | 39.92 (32.35, 49.26) | 6.06 (3.15, 11.66) | 6.11 (2.12, 17.61) |

*CI* confidence interval.

[a]The average annual change is calculated using a linear regression model, with log-transformed consumption in DDD/TID as the dependent variable and year as the independent variable. The average annual change was expressed as average annual percentage change, calculated by [exp(the coefficient of the year variable) – 1] × 100%.

stratified the sales data based on country income levels (i.e., lower-middle income, upper-middle income, and high income according to the 2018 World Bank income classification)[68] to investigate how consumption trends vary with country income levels. Five countries, including Bosnia and Herzegovina, Kazakhstan, Netherlands, Serbia and Thailand, recorded no sales data for gabapentinoids before 2011. Although this reflects that no sales were made, we conducted two sets of sensitivity analyses on gabapentinoid consumption rates and trends, with one removing the sales data of the above countries and the other using only retail data to test the robustness of our results against potential missing data. To investigate whether imputation of missing strength of the product affected the finding of the study, we performed another set of sensitivity analysis by removing products that have missing strength. A 95% CI not overlapping with the null was considered statistically significant. All analyses were conducted using Statistical Analysis System (SAS) v9.4 (SAS Institute, Cary, NC, USA) and R Foundation for Statistical Computing version 3.6.0 (Vienna, Austria).

**Reporting summary**

Further information on research design is available in the Nature Portfolio Reporting Summary linked to this article.

## Data availability

The MIDAS data from IQVIA are available under restricted access for licensing reasons, access can be obtained by entering into additional licensing agreement with IQVIA. The raw MIDAS data were protected and are not publicly available due to data protection agreement with IQVIA. With additional data use agreement and permission from IQVIA, MIDAS data will be made available from the corresponding authors upon request. Source data of tables and figures presented are provided with this paper. Data requisition can be made by emailing the corresponding authors I.C.K.W. and K.K.C.M. at wongick@hku.hk and kenneth.man@ucl.ac.uk respectively. Response to request for MIDAS data will be provided within 1 month. Source data are provided with this paper.

## Code availability

R codes adopted in this study have been made available on GitHub repository at https://github.com/adrienneylc/Gabapentinoids.

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

## Acknowledgements
We thank IQVIA for their assistance and information regarding the use of IQVIA-MIDAS data.

## Author contributions
A.Y.L.C., A.S.C.Y., K.K.C.M. and I.C.K.W. had full access to the aggregate analysis data in the study and take responsibility for the integrity of the data and the accuracy of the data analysis. I.C.K.W. and K.K.C.M. were responsible for the study concept, and I.C.K.W., K.K.C.M., A.Y.L.C. and A.S.C.Y. were responsible for the study design. A.Y.L.C. and A.S.C.Y. did the statistical analysis. A.Y.L.C., A.S.C.Y., K.K.C.M. and I.C.K.W. drafted the manuscript. A.Y.L.C., A.S.C.Y., D.H.T.T., W.C.Y.L., Y.H.J., Y.H., D.P.J.O., J.F.H., F.M.C.B., E.C.C.L., L.W., K.T., I.C.K.W. and K.K.C.M. critically revised the manuscript for important intellectual content. L.W., I.C.K.W. and K.K.C.M were responsible for database acquisition.

## Competing interests
A.Y.L.C. is supported by the AIR@innoHK programme of the Hong Kong Innovation and Technology Commission. D.P.J.O and J.F.H are supported by the University College London Hospitals NIHR Biomedical Research Centre and the NIHR North Thames Applied Research Collaboration. I.C.K.W. received research funding outside the submitted work from Amgen, Bristol-Myers Squibb, Pfizer, Janssen, Bayer, GSK, Novartis, Takeda, the Hong Kong Research Grants Council, and the Hong Kong Health and Medical Research Fund, National Institute for Health Research in England, European Commission, National Health and Medical Research Council in Australia. He is also a non-executive director of Jacobson Pharma Corporation Limited in Hong Kong and a consultant to the World Health Organization. K.K.C.M. reports grants from the CW Maplethorpe Fellowship, the European Union Horizon 2020, the UK National Institute of Health Research and the Hong Kong Research Grant Council, Hong Kong Innovation and Technology Commission, and reports personal fees from IQVIA, unrelated to the submitted work. The remaining authors declare no competing interests. All funders had no role in study design, data collection, data analysis, data interpretation, or writing of the report. The views expressed in this article are those of the authors and not necessarily those of the NHS, the NIHR, or the Department of Health and Social Care.
