## [Peer Review File · Nature Communications]

nature portfolio

Peer Review FileReviewer comments, first round

Reviewer #1 (Remarks to the Author):

I read with interest the paper "Gabapentinoid consumption in 65 countries and regions from 2008 to 2018: a longitudinal study". The purpose of this study was to evaluate the global trends in gabapentinoid consumption between 2008-2018 to examine the need for any interventional policies to prevent gabapentinoid abuse and misuse. Major and minor revisions are needed as itemized below:

Title

The title can be better rephrased as it is not clear what is meant with regions and authors may consider specifying what study design was employed rather than indicating that it is a longitudinal study.

Background

Scientific background and rationale for the investigation is reported along with specific objective, although safety risk associated with gabapentinoid use/abuse/misuse should be emphasized describing some more into details the cited studies that evaluated the safety profile associated with gabapentinoid abuse/misuse.

In addition, all the studies reported were related to US and UK, additional information on gabapentinoids use in other part of the world or on potential predictors (e.g. socioeconomic status, healthcare system organization, availability of drugs as generic, etc.) of use which may differ across countries should be described.

Methods

Data source: authors mention that "The MIDAS database has been validated against external data sources and used as a proxy to evaluate multinational consumption of different classes of medications." Was validation already performed for all the countries participating in this study? As the study included 65 countries/regions, authors should clarify if the accuracy and coverage of this data source with respect to gabapentinoids use is expected to be homogeneously high across all participating countries. In particular, authors should reassure about the fact that observed lower use of gabapentinoids in Central Asia is not be ascribed to lower coverage of the data source.

Data inclusion

In order to have a better understanding of the comparison of the sales data across Countries, authors should provide more information on the availability of the study drugs across participating countries, in addition to what is reported in table S2. What was indication of use approved in different countries for each drug? What about reimbursement status and presence of generic? Was the cost of the drugs charged directly to the citizens in special conditions? Was there any major guideline potentially having impact on the drug use?

Statistical analysis (Page 8, line 158): since the drugs included in the study were gabapentin, pregabalin and gabapentin enacarbil as monotherapy, the authors should specify why they mentioned combination product as reported in the following statement "DDD for combination products was converted from a standard unit (defined as a single tablet, capsule, or ampoule/vial or 5 mL oral solution/suspension), formulation, with their respective drug ingredients....".

Analyses for which results have been reported in Figure 1, Figure 3, Table S4-6 and Figure S1 should be described in details in the methods section.

Results

I suggest keeping Figure S1 in the main paper as it is very informative and the difference in terms of consumption for gabapentin, pregabalin and gabapentin enacarbil in every country/region is

described both in results (lines 226-239) and discussion (lines: 288-314).

Discussion and limitations

Key results are well summarized, and limitations are discussed.

The reasons why Puerto Rico is the country with the highest consumption of gabapentoid in general in 2018 (Table 1), and gabapentin in particular (Figure S1), also need to be discussed.

Conclusion

I suggest being more cautious in providing final recommendations in the conclusion, based in the study findings. In particular, the use of sales data is to be intended as only for exploratory analyses and in no way can directly inform healthcare policy interventions as key information are missing such as indication of use as also acknowledged by the authors. As such I suggest reevaluate the implications of the study findings

Other minor revisions:

Results

Page 9 (line 184): Remove the word "regions".

Page 10 (lines 186-188): Add the reference to Table 1.

Page 10 (line 192): Add the word "Average" at the beginning of the sentence.

Page 12 (lines: 227-239): Add the reference to Figure S1.

Please move lines 226-227 ("Gabapentin and pregabalin were available in all studied countries whereas gabapentin enacarbil, a prodrug of gabapentin, was only sold in the US, Puerto Rico, and Japan (Table S2).") in the section where Table S2 was commented (line 189)

Discussion

Page 16 (line 329): Remove the word "65"

Reviewer #2 (Remarks to the Author):

Thank you very much for the opportunity to review the manuscript. This is a very timely contribution regarding a topic the importance of which will likely only continue to grow. I would like to commend the authors for dedicating the time to obtain and analyze the data, it will be very useful to have this information available in an academic publication. The interpretation of data pooled from many different national contexts can be tricky (and I assume for some readers country-level information will be of great interest) but the overall large picture messages of international trends and differences are helpful indeed. I am primarily familiar with the North American context and the thrust of the results offered here is in line with observations made by others, even if the underlying data may differ.

The analysis uses presumably one of the best-available international data and is largely descriptive in character. There are a few areas where I think additional detail on data and methods would be helpful. Specifically, I had five main observations pertaining to the analyses and their presentation that I wanted to raise for the authors' consideration.

1) I think it would be helpful to offer a commentary on the fit of the regression models run by the team, their susceptibility to extreme values, and justification of the specification used for the main model. There were two results that particularly caught my attention. First, the estimated average global annual increase of 23.14% would imply a more than eightfold increase over the study period, which is much more than the observed difference between the estimated 2008 and 2018 values. Second, the sensitivity analysis consisting of removing five countries resulted in a nearly 30% downward revision of the estimate of average annual change, which is a notable difference.

2) I appreciated the stratification of countries into income groups, taking into account the inherent limitation that it groups together many disparate national contexts. Within the high-income group, there appears to be a split between very high prescribing regions and comparatively low ones, such as Latin America and Eastern Asia. I was wondering if introducing a subgroup of the high-income group, e.g., OECD countries, may also be instructive. Not least because those are likely countries where the risks and concerns regarding polydrug use and other issues discussed in the paper are likely most pronounced.

3) It is regrettable that data on hospital pharmacies were not available for all countries. I think it would be helpful to also offer an analysis of retail-only data (perhaps as a supplementary analysis) to see how much, if at all, the differences in data availability affect the observed results, including differences across regions and income groups.

4) Relatedly, would the team be in a position to comment on whether the imputations done by IQVIA (to account for incomplete coverage) and by the team (to account for missing data on formulation and strength) risk biasing the study in a particular direction?

5) I was wondering if it would be useful to also split the study period into two halves to see if there were any changes in observed trends. One rationale for doing so, consistent with the themes covered in the discussion section (particularly row 273 and onwards), would be to explore any potential response to increasingly tightening opioid prescribing practices. This may be primarily applicable to the North American context but perhaps also elsewhere?

In addition, I had a few questions in instances where I may not have properly understood.

- Row 268: "We cautiously recommend revisiting the appropriateness of the prescriptions." Could you explain what that would entail in practice? Is that a retrospective research suggestion or a clinical practice recommendation?

- Row 275: "In some of the studied countries, the less stringent requirements for prescription and product storage compared to opioids and benzodiazepines, might have increased the ease of access to gabapentinoids for prescribers and patients." Could you elaborate on the storage requirement aspect and why that would be a factor? I am not immediately clear on how that would affect prescribers' decisionmaking. It also does not affect the ease of access to gabapentinoids for prescribers, I'd think?

- Paragraph beginning on row 273 and continuing on the next page: I think there are actually two potentially distinct points here, correct? The first is risks stemming from the co-prescribing of some combination of gabapentinoids, benzodiazepines, and opioids, and the other is the risks of misuse stemming from patients switching from opioids to gabapentinoids.

- Row 326: I am not sure why it is necessary to have individual-level data to assess risks of misuse and population-level data would not be sufficient for the type of analysis suggested here

- Row 327: I completely take the authors' point that only licit sales are included in the data but not sure I fully follow the implications. My understanding is that there is no meaningful illegal manufacturing of gabapentinoids and diversion of gabapentinoids primarily occurs after a prescription has been issued/sale has been made. Assuming this is correct, this would mean that even medications that would subsequently get diverted and sold in illicit markets would be captured by IQVIA data. In that case, from the perspective of this manuscript I don't think it matters whether the prescribed drugs were consumed by the original recipient or someone else or whether it was for medical reasons. In other words, the bigger question would be whether the prescription was consumed or not, rather than by whom and why. Or did I misunderstand and there is a concern that a notable share of gabapentinoid consumption is enabled by diversion from the healthcare system without involving any record (e.g., theft)? I suppose international trafficking of gabapentinoids could be another data issue, but again I am not aware that this would be a significant phenomenon?

Lastly, I would suggest a few language edits:

- I understand that "abuse potential" is still language used by some stakeholders but in other instances, I would suggest avoiding the term "abuse." For instance rows 115-116, 274, 284, 326.

- Row 186: would suggest changing to "annual percentage change of gabapentinoid consumption"

- Row 278 and 341: "dependency": elsewhere in the manuscript and more often "dependence" is used

- Row 308: would suggest changing "was potentially because of" to "may have been caused by"

- Row 329: two sentences appear to have been mixed up

Response letter to reviewers and editors

REVIEWER COMMENTS

Reviewer #1 (Remarks to the Author):

I read with interest the paper “Gabapentinoid consumption in 65 countries and regions from 2008 to 2018: a longitudinal study”. The purpose of this study was to evaluate the global trends in gabapentinoid consumption between 2008-2018 to examine the need for any interventional policies to prevent gabapentinoid abuse and misuse. Major and minor revisions are needed as itemized below:

Response: We thank the reviewer for their constructive comments.

Title

The title can be better rephrased as it is not clear what is meant with regions and authors may consider specifying what study design was employed rather than indicating that it is a longitudinal study.

Response: Thank you for the comment. We are referring “regions” to geographical locations in this paper, and were categorised into “65 countries and regions” as stated below.

“The included countries were divided ... according to United Nations’ (UN) “Standard Country or Area Codes for Statistical Use”⁶². (page 14, line 307-312)

This classification method is in accordance with the United Nations for statistical purposes and does not imply the expression of any opinion concerning the legal status of any country, territory, city or area or of its authorities, or concerning the delimitation of its frontiers or boundaries. The term “regions” was used to cover geographical locations such as Puerto Rico or Taiwan, which are not classified as a country by some international organisations.

We updated the title to

“Gabapentinoid consumption in 65 countries and regions from 2008 to 2018 a longitudinal trend study”

to reflect the study design on page 1 line 2-3. We believe “longitudinal trend study” is the most appropriate phrase to reflect our study design due to the ecological nature of the study which uses country-level pharmaceutical sales data.

The wording used in the current title has also been used in previous major publications:

1. Brauer R, Alfageh B, Blais JE, Chan EW, Chui CSL, Hayes JF, et al. Psychotropic medicine consumption in 65 countries and regions, 2008-19: a longitudinal study. *Lancet Psychiatry*. 2021;8(12):1071-82.
2. Ju C, Wei L, Man KKC, Wang Z, Ma TT, Chan AYL, et al. Global, regional, and national trends in opioid analgesic consumption from 2015 to 2019: a longitudinal study. *Lancet Public Health*. 2022;7(4):e335-e46.
3. Chan AY, Ma TT, Lau WC, Ip P, Coghill D, Gao L, Jani YH, Hsia Y, Wei L, Taxis K, Simonoff E. Attention-deficit/hyperactivity disorder medication consumption in 64 countries and regions from 2015 to 2019: A longitudinal study. *eClinicalMedicine*. 2023 Mar 20:101780.

Background

Scientific background and rationale for the investigation is reported along with specific objective, although safety risk associated with gabapentinoid use/abuse/misuse should be emphasized describing some more into details the cited studies that evaluated the safety profile associated with

gabapentinoid abuse/misuse.

Response: Thank you for the comment. We have revised the introduction section to emphasise the scale and seriousness of gabapentinoid misuse presented in other studies.

“The prominent use of gabapentinoids has raised concerns about their potential misuse, which can lead to eventual episodes of hospitalisation or mortality, especially for patients with a history of substance abuse and psychiatric comorbidities¹⁶⁻¹⁸. A UK study has shown that gabapentinoid-related overdose fatalities have also increased substantially in recent years, and 79% of them also involve the use of opioids¹⁹. Gabapentin was also reported to be the most misused non-controlled medication in a prison setting in the US²⁰. The misuse of gabapentinoids can be explained by not only their euphoric and relaxation effects but also their potential reduction of withdrawal effects of other drugs^{18,21,22}.” (page 4-5, Line 81 to 89)

In addition, all the studies reported were related to US and UK, additional information on gabapentinoids use in other part of the world or on potential predictors (e.g. socioeconomic status, healthcare system organization, availability of drugs as generic, etc.) of use which may differ across countries should be described.

Response: Thank you for the comment. We identified three more studies from Sweden, Australia and Taiwan, and all indicated an increasing trend of gabapentinoid prescriptions related to pain management or off-label indications, which align with the studies conducted in the UK and US.

“Similar prescribing trends for pain management and off-label indications were also observed in other countries/ regions such as Sweden, Australia and Taiwan, which have different demographic composition and healthcare systems when compared to the UK and US¹³⁻¹⁵.” (page 4, line 78-81)

While we did not find any studies investigating the potential predictors of gabapentinoid use, we agree with the reviewer that information about potential predictors of use is important and added a call for further studies on this topic in the discussion section,

“In view of the increasing concerns over their dependence and misuse potential, further studies are also needed to monitor the safety and appropriateness of gabapentinoid use and to investigate the potential predictors of the increase in gabapentinoid use.” (page 13, line 270-272)

Methods

Data source: authors mention that “The MIDAS database has been validated against external data sources and used as a proxy to evaluate multinational consumption of different classes of medications.” Was validation already performed for all the countries participating in this study? As the study included 65 countries/regions, authors should clarify if the accuracy and coverage of this data source with respect to gabapentinoids use is expected to be homogeneously high across all participating countries. In particular, authors should reassure about the fact that observed lower use of gabapentinoids in Central Asia is not be ascribed to lower coverage of the data source.

Response: We thank the reviewer for raising this concern. Validation studies are conducted for each product registered in individual countries, allowing comparisons of cross-national purchases. IQVIA performs an annual internal statistical program, Accuracy and Timeliness Statistics (ACTS), a quality assurance program that validates purchasing data based on accuracy and timeliness through alternate sources, including manufacturers and pharmaceutical companies.

The market coverage in Central Asia (Kazakhstan) has sales data from retail and hospitals with a market coverage of 100% as reported in previous publications. Furthermore, we also cited a previous paper showing the market coverage of each country in the MIDAS data,

“For countries where the MIDAS database did not have 100% sector coverage, adjustments were made by IQVIA to estimate the total sales volume based on knowledge of the market share of participating wholesalers and retail or hospital pharmacies^{54,55}.” (page 14, line 292-295)

References:

1. Cook MN. Estimating national drug consumption using data at different points in the pharmaceutical supply chain. *Pharmacoepidemiol Drug Saf* 2006; 15(10): 754-7.
2. Ju C, Wei L, Man KKC, et al. Global, regional, and national trends in opioid analgesic consumption from 2015 to 2019: a longitudinal study. *Lancet Public Health* 2022; 7(4): e335-e46.

We have also conducted an additional sensitivity analysis to evaluate the robustness of our study results and conclusion against the impact of market coverage by only including Retail data which demonstrated similar results as in the main analyses in terms of gabapentinoid consumption level and annual average percentage change. (page 7, line 136-137; page 16, line 339-342; Supplementary Table 4)

Data inclusion

In order to have a better understanding of the comparison of the sales data across Countries, authors should provide more information on the availability of the study drugs across participating countries, in addition to what is reported in table S2. What was indication of use approved in different countries for each drug? What about reimbursement status and presence of generic? Was the cost of the drugs charged directly to the citizens in special conditions? Was there any major guideline potentially having impact on the drug use?

Response: Thank you for the comment. We would like to clarify that the original purpose of this study is to look at the trend of gabapentinoid use in different countries. We are fully aware that there could have differences in not only the factors that the reviewer had mentioned but also in the clinical practices, culture, attitude towards the use of medication, etc. However, it is difficult to understand the sales across countries without quantifying the trend in the very first place and therefore we decided to focus on the trend in the current study. We agree with the reviewer that these factors (approved indication of use, reimbursement status, guidelines in different countries, and more) could affect the consumption of gabapentinoids and the investigation into the effect of these effects is of timely relevance and would be great to have further study to look into this issue. We have added this to the discussion section,

“In view of the increasing concerns over their dependence and misuse potential, further studies are also needed to monitor the safety and appropriateness of gabapentinoid use and to investigate the potential predictors of the increase in gabapentinoid use.” (page 13, line 270-272)

Statistical analysis (Page 8, line 158): since the drugs included in the study were gabapentin, pregabalin and gabapentin enacarbil as monotherapy, the authors should specify why they mentioned combination product as reported in the following statement “DDD for combination products was converted from a standard unit (defined as a single tablet, capsule, or ampoule/vial or 5 mL oral solution/suspension), formulation, with their respective drug ingredients....”.

Response: We appreciate the comments shared by the reviewer. We wish to first clarify that we are not sure if the medications were used as monotherapy or not as that piece of information is not available in sales data. While DDD was designed for single-molecule product or monotherapy, some of the gabapentinoid products included in the dataset were in combination with other chemicals as gabapentinoid may sometimes be formulated with other active ingredients. Therefore, similar to previous studies^{1,2}, we have to estimate the strength and DDD count for some gabapentinoid products in this study. To account for the consumption of gabapentinoids from these combination products, DDD count of them was converted from a standard unit as explained in the following sentence,

“DDD is the assumed average maintenance dose per day for a drug used for its main indication in adults and was only available for single-molecule products⁶⁴. As such, DDD count for combination products was converted from a standard unit (defined as a single tablet, capsule, or ampoule/vial or 5 mL oral solution/ suspension), formulation, with their respective drug ingredients mapped to the ATC/DDD Index developed by the World Health Organisation (WHO) Collaborating Centre for Drug Statistics Methodology (Supplementary Table 1)⁶⁵. Where the strength or formulation of the product was missing, they were imputed based on the respective information of the most-sold product of the same drug⁵³.” (page 15, line 316-324)

Reference:

1. Ju C, Wei L, Man KKC, Wang Z, Ma TT, Chan AYL, et al. Global, regional, and national trends in opioid analgesic consumption from 2015 to 2019: a longitudinal study. *Lancet Public Health*. 2022;7(4):e335-e46.
2. Chan AY, Ma TT, Lau WC, Ip P, Coghill D, Gao L, Jani YH, Hsia Y, Wei L, Taxis K, Simonoff E. Attention-deficit/hyperactivity disorder medication consumption in 64 countries and regions from 2015 to 2019: A longitudinal study. *eClinicalMedicine*. 2023 Mar 20:101780.

Analyses for which results have been reported in Figure 1, Figure 3, Table S4-6 and Figure S1 should be described in details in the methods section.

Response: Thank you for the comment. The analytical methods were reported according to the RECORD-PE statement checklist of items, extended from the STROBE and RECORD statements.

Reference:

Langan SM, Schmidt SA, Wing K, et al. The reporting of studies conducted using observational routinely collected health data statement for pharmacoepidemiology (RECORD-PE). *BMJ*. 2018;363:k3532. Published 2018 Nov 14. doi:10.1136/bmj.k3532

Results

I suggest keeping Figure S1 in the main paper as it is very informative and the difference in terms of consumption for gabapentin, pregabalin and gabapentin enacarbil in every country/region is described both in results (lines 226-239) and discussion (lines: 288-314).

Response: We agree with the reviewer and will place Figure S1 in the main paper. It is now numbered as Figure 2 in the main paper.

Discussion and limitations

Key results are well summarized, and limitations are discussed.

The reasons why Puerto Rico is the country with the highest consumption of gabapentinoid in general in 2018 (Table 1), and gabapentin in particular (Figure S1), also need to be discussed.

Response: Thank you for the comment. As shown in Table 1 and Figure 2 (previously Figure S1), both

Puerto Rico and the US share a similar trend in gabapentinoid consumption, with gabapentin taking up the majority of the increase in use. We are not sure about the exact reason, but it is likely be due to the fact that Puerto Rico is a territory of the US. Many of its healthcare professionals are trained in the US and may share similar clinical beliefs as their US counterparts. Our results showed that Puerto Rico and the US (ranked 1st and 2nd) share similar DDD/TID in 2018 which may be because of their similar healthcare practice. The similarity between Puerto Rico and the US can also be spotted in the use of medication as gabapentin is more commonly used when compared to pregabalin, potentially with the similar rationale as the US (page 11, Line 222-236).

Conclusion

I suggest being more cautious in providing final recommendations in the conclusion, based in the study findings. In particular, the use of sales data is to be intended as only for exploratory analyses and in no way can directly inform healthcare policy interventions as key information are missing such as indication of use as also acknowledged by the authors. As such I suggest re-evaluate the implications of the study findings

Response: Thank you for the comment. We strongly agree with the reviewer and our study did not intend for readers to directly take its results for policy change. Indeed, we only recommend a review of current policies based on the observed trends. We have slightly revised the conclusion to make this clear.

“Against the background of this evidence of increasing use, considering both the abuse potential and the mixed evidence for off-label use, further studies are warranted to investigate the implications behind the increase in consumption and if there is a case for international and national regulatory bodies to review existing treatment guidelines and public health policy relating to gabapentinoids.” (page 13, line 278-282)

Other minor revisions:

Results

Page 9 (line 184): Remove the word “regions”.

Response: We included the term “regions” to cover geographical locations such as Puerto Rico or Taiwan, which are not classified as a country by some international organisations. Detailed explanation is stated above.

Page 10 (lines 186-188): Add the reference to Table 1.

Response: We have made the change accordingly.

“The average annual percentage change of gabapentinoid consumption was +17.20% (95%CI, +15.52% to +18.91%), from 4.17 DDD/TID (95%CI, 2.99 to 5.81) in 2008 to 18.26 DDD/TID (95%CI, 13.54 to 24.63) in 2018 (Table 1).” (page 6, line 108-110)

Page 10 (line 192): Add the word “Average” at the beginning of the sentence.

Response: We have made the change accordingly.

“Average annual increase in consumption was the highest in Northern Africa (+35.91%; 95%CI, +26.17% to +46.41%), followed by Eastern Asia (+28.51%; 95%CI, +18.86% to +38.94%), Eastern Europe (+23.77%; 95%CI, +17.06% to +30.86%), Central Asia, (+20.45%; 95%CI, -0.53% to +45.85%), Oceania (+19.89%; 95%CI, +13.70% to +26.43%), Western Asia (+17.57%; 95%CI, +10.53% to +25.06%), Southern Asia (+15.56%; 95%CI, +12.19% to +19.03%), Southern Europe (+14.91%; 95%CI, +10.92% to +19.05%), Northern Europe (+14.78%; 95%CI, +12.34% to +17.27%), South-eastern Asia (+14.70%; 95%CI, +9.50% to

+20.04%), Central and Southern America and the Caribbean (+12.92%; 95%CI, +9.55% to +16.39%), Southern Africa (+12.42%; 95%CI, +6.35% to +18.85%), Northern America (+9.04%; 95%CI, +6.82% to +11.32%), and Western Europe (+8.15%; 95%CI, +6.64% to +9.68%).” (page 6, line 116-126)

Page 12 (lines: 227-239): Add the reference to Figure S1.

Response: We have made the change accordingly.

“At country/territory-level, Puerto Rico had the highest consumption of gabapentin (139.25 DDD/TID; 95%CI 139.18 to 139.32), Japan had the highest consumption of gabapentin enacarbil (0.63 DDD/TID; 95%CI, 0.63 to 0.64), and Australia had the highest consumption of pregabalin (87.38 DDD/TID; 95%CI, 87.36 to 87.40; Figure 2).” (page 8, line 158-162)

Please move lines 226-227 (“Gabapentin and pregabalin were available in all studied countries whereas gabapentin enacarbil, a prodrug of gabapentin, was only sold in the US, Puerto Rico, and Japan (Table S2).”) in the section where Table S2 was commented (line 189)

Response: We have made the change accordingly.

“The characteristics of included countries and the availability of different gabapentinoids sold were presented in Supplementary Table 2. Gabapentin and pregabalin were available in all studied countries whereas gabapentin enacarbil, a prodrug of gabapentin, was only sold in the US, Puerto Rico, and Japan.” (page 6, line 110-113)

Discussion

Page 16 (line 329): Remove the word “65”

Response: We have made the change accordingly.

“Third, although 70% of the world population were included in our study, the findings do not necessarily apply to the countries that are not included in the dataset.” (page 12, line 251-253)

Reviewer #2 (Remarks to the Author):

Thank you very much for the opportunity to review the manuscript. This is a very timely contribution regarding a topic the importance of which will likely only continue to grow. I would like to commend the authors for dedicating the time to obtain and analyze the data, it will be very useful to have this information available in an academic publication. The interpretation of data pooled from many different national contexts can be tricky (and I assume for some readers country-level information will be of great interest) but the overall large picture messages of international trends and differences are helpful indeed. I am primarily familiar with the North American context and the thrust of the results offered here is in line with observations made by others, even if the underlying data may differ.

The analysis uses presumably one of the best-available international data and is largely descriptive in character. There are a few areas where I think additional detail on data and methods would be helpful. Specifically, I had five main observations pertaining to the analyses and their presentation that I wanted to raise for the authors' consideration.

Response: We thank the reviewer for the constructive comments.

1) I think it would be helpful to offer a commentary on the fit of the regression models run by the team, their susceptibility to extreme values, and justification of the specification used for the main model.

Response: We thank the reviewer for raising this concern which we shared a similar concern when planning our study. In our model, we have log-transformed the raw data on consumption measures to account for potential extreme values. Further, upon examining the distribution of the data, graphically represented in Figure 1, we did not find any major concerns regarding our statistical models in the analyses. More importantly, the purpose of the statistical models is aimed to describe the observed trends and does not intend for the results to be extrapolated for predictive purposes, formal diagnostic statistics were not computed on the model goodness-of-fit.

There were two results that particularly caught my attention. First, the estimated average global annual increase of 23.14% would imply a more than eightfold increase over the study period, which is much more than the observed difference between the estimated 2008 and 2018 values. Second, the sensitivity analysis consisting of removing five countries resulted in a nearly 30% downward revision of the estimate of average annual change, which is a notable difference.

Response: We apologise for the mistake here. After rechecking all the figures in the manuscript and all tables, we found that average global annual increase of gabapentinoids was mixed up with the one of pregabalin. We have corrected the mistakes accordingly. The multinational average annual percentage change of gabapentinoid consumption should be +17.20% (95%CI, +15.52% to +18.91%), and the value for pregabalin should be +23.14% (95%CI, +20.07% to +26.28%).

However, we wish to clarify that the estimated average annual percentage change was based on the multi-level linear mixed model that addressed the heterogeneity across the countries/regions. Using the pooled estimates in 2008 and 2018 will not end up with the same number generated from the regression model. Please see the figure below for the data points and the fitted regression line for further information.

Secondly, we agree that there is a decrease in average annual increase of gabapentinoid. However, after correcting the mistake as mentioned in the previous comment, the absolute reduction is now only approximately 0.5% (from 17.20% to 16.68%), the average annual increase remained high, thus the downward revision did not affect the conclusion of our study.

Furthermore, after consulting the data provider (IQVIA), “zero” units sold in the data is more likely to be the case in that country in that particular year. Therefore, we included these data in our main analysis. We revised the description as follows:

“Although this reflects that no sales were made, we conducted two sets of sensitivity analyses on gabapentinoid consumption rates and trends, with one removing the sales data of the above countries and the other using only retail data to test the robustness of our results against potential missing data.” (page 16, line 339-342)

2) I appreciated the stratification of countries into income groups, taking into account the inherent limitation that it groups together many disparate national contexts. Within the high-income group, there appears to be a split between very high prescribing regions and comparatively low ones, such as Latin America and Eastern Asia. I was wondering if introducing a subgroup of the high-income group, e.g., OECD countries, may also be instructive. Not least because those are likely countries where the risks and concerns regarding polydrug use and other issues discussed in the paper are likely most pronounced.

Response: Thank you for the comment. We agree with the reviewer that country-income levels may not be the defining factors behind different consumption levels of gabapentinoids and call for future studies looking into these factors in the discussion section.

“In view of the increasing concerns over their dependence and misuse potential, further studies are also needed to monitor the safety and appropriateness of gabapentinoid use and to investigate the potential predictors of the increase in gabapentinoid use.” (page 13, line 270-272)

3) It is regrettable that data on hospital pharmacies were not available for all countries. I think it would be helpful to also offer an analysis of retail-only data (perhaps as a supplementary analysis) to see how much, if at all, the differences in data availability affect the observed results, including differences across regions and income groups.

Response: Thanks for the comment. We agree with the reviewer that it is important to examine the effect of data availability on our estimates. We conducted an additional sensitivity analysis using only retail data and found little impact on our overall estimates but only affected estimates for certain countries. The multinational average annual percentage change of gabapentinoid consumption with retail data only is +17.49% (95%CI, +15.65% to +19.37%), which closely resembles the value in the primary analysis: +17.20% (95%CI, +15.52% to +18.91%).

We added the following,

“Although this reflects that no sales were made, we conducted two sets of sensitivity analyses on gabapentinoid consumption rates and trends, with one removing the sales data of the above countries and the other using only retail data to test the robustness of our results against potential missing data.” (page 16, line 339-342)

And added to our discussion,

“Fourth, gabapentinoid consumption could be underestimated in countries without 100% market coverage despite adjustments made to project the total consumption, especially in countries that did not have hospital coverage. However, total pharmaceutical market coverage in most countries was greater than 80%. This may affect the estimation of consumption levels but unlikely to influence the estimation of trends since it is a relative measure. Our sensitivity analysis using only retail data showed that the lack of hospital coverage only affected the estimates for individual countries and did not significantly affect the multinational or regional consumption levels.” (page 12, line 253-260)

4) Relatedly, would the team be in a position to comment on whether the imputations done by IQVIA (to account for incomplete coverage) and by the team (to account for missing data on formulation and strength) risk biasing the study in a particular direction?

Response: Thank you for the comment. We believe, as the same imputation approach was applied within each country by IQVIA for incomplete coverage, it may introduce systematic bias to our study but should not affect the study conclusion as the same bias should exist throughout the study period. In addition, the main outcome of the study, annual average percentage change of gabapentinoid consumption, is a relative measure by which the systematic bias would be cancelled out in the estimation.

Imputation was conducted by us to account for missing strength information. As this is a descriptive study, we believe this is a conservative approach to estimate the strength with the most common formulation of the product. Furthermore, missing strength information is only observed in combination products, i.e. product with more than one active drug ingredients. The same approach

for identifying the strengths for these products as used in other studies^{1,2}. Therefore, the bias induced should be minimal.

Limitation of this approach is that it may affect the estimation of consumption level. However, we are not in the position to comment on the direction of bias as median imputation was used and the direction is related to the actual product. We therefore conducted a sensitivity analysis to quantify the potential impact of the imputed strengths. After including only single-molecule products, the pooled multinational consumption levels were 4.11 DDD/TID (95%CI, 2.99 to 5.66) in 2008 and 17.84 DDD/TID (95%CI, 13.39 to 23.77) in 2018, which are comparable to our main analysis: 4.17 DDD/TID (95%CI: 2.99 to 5.81) in 2008 and 18.26 DDD/TID (95%CI, 13.54 to 24.63) in 2018.

Reference:

1. Ju C, Wei L, Man KKC, Wang Z, Ma TT, Chan AYL, et al. Global, regional, and national trends in opioid analgesic consumption from 2015 to 2019: a longitudinal study. *Lancet Public Health*. 2022;7(4):e335-e46.
2. Chan AY, Ma TT, Lau WC, Ip P, Coghill D, Gao L, Jani YH, Hsia Y, Wei L, Taxis K, Simonoff E. Attention-deficit/hyperactivity disorder medication consumption in 64 countries and regions from 2015 to 2019: A longitudinal study. *eClinicalMedicine*. 2023 Mar 20:101780.

5) I was wondering if it would be useful to also split the study period into two halves to see if there were any changes in observed trends. One rationale for doing so, consistent with the themes covered in the discussion section (particularly row 273 and onwards), would be to explore any potential response to increasingly tightening opioid prescribing practices. This may be primarily applicable to the North American context but perhaps also elsewhere?

Response: We thank the reviewer for the suggestion and indeed it would be interesting to explore the impact of particular landmark events on the trend changes of gabapentinoid use. However, as illustrated in previous publications, the trends of benzodiazepine and opioid use differ from country to country and they are influenced by country-specific events, thus no single timepoint could be identified for analyses.

References:

1. Brauer R, Alfageh B, Blais JE, Chan EW, Chui CSL, Hayes JF, et al. Psychotropic medicine consumption in 65 countries and regions, 2008-19: a longitudinal study. *Lancet Psychiatry*. 2021;8(12):1071-82.
2. Ju C, Wei L, Man KKC, Wang Z, Ma TT, Chan AYL, et al. Global, regional, and national trends in opioid analgesic consumption from 2015 to 2019: a longitudinal study. *Lancet Public Health*. 2022;7(4):e335-e46.

In addition, I had a few questions in instances where I may not have properly understood.

- Row 268: "We cautiously recommend revisiting the appropriateness of the prescriptions." Could you explain what that would entail in practice? Is that a retrospective research suggestion or a clinical practice recommendation?

Response: Thank you for the comment. This sentence is a clinical recommendation that echoes the suggestions made in other studies referenced. As prescribing of gabapentinoids is often related to off-label use, which are based on previous clinical experience with limited evidence or sometimes influenced by misleading marketing campaigns from the manufacturers, hence, clinicians should reconsider the necessity and appropriateness of prescribing gabapentinoids for their patients, especially for off-label uses.

- Row 275: "In some of the studied countries, the less stringent requirements for prescription and product storage compared to opioids and benzodiazepines, might have increased the ease of access to gabapentinoids for prescribers and patients." Could you elaborate on the storage requirement aspect and why that would be a factor? I am not immediately clear on how that would affect prescribers' decisionmaking. It also does not affect the ease of access to gabapentinoids for prescribers, I'd think?

Response: Thanks for the comment. A more stringent prescription requirements may influence the consumption of individual medication. For instance, in the UK, if a drug is classified as a controlled drug (CD), the prescription will only be valid for 28 days, which is much shorter when compared to the 6 months validity period of other prescription-only medications. A shorter validity of the prescription will lead to a more frequent review of the treatment regimen by the prescriber, which may in turn prompt a timely termination of unwarranted medications. Prescription of CD can also only be issued by a limited types of prescribers which restrict the access to these types of medications.

The Department of Health and Social Care in the UK has also recommended the prescriptions of CD should limit the quantity for up to 30 days.

However, we agree that product storage may not have a direct impact to the consumption of gabapentinoids or the decision-making process by the prescribers. Hence, the sentence has now been amended to

"In some of the studied countries, the less stringent requirements for prescription compared to opioids and benzodiazepines, might have increased the ease of access to gabapentinoids for prescribers and patients." (page 10, line 197-199)

- Paragraph beginning on row 273 and continuing on the next page: I think there are actually two potentially distinct points here, correct? The first is risks stemming from the co-prescribing of some combination of gabapentinoids, benzodiazepines, and opioids, and the other is the risks of misuse stemming from patients switching from opioids to gabapentinoids.

Response: We thank the reviewer for the clarification. The sentences

"Another potential reason for the increase in gabapentinoid consumption is the increasing concerns over misuse of opioids and benzodiazepines^{34-36,37}. Gabapentinoids have often been perceived as safer alternatives^{1,34-37}." (page 10, line 195-197)

was only intended to illustrate the reasons behind increase in gabapentinoid consumption and not to discuss the risks arising from co-prescribing with or switching to gabapentinoids.

- Row 326: I am not sure why it is necessary to have individual-level data to assess risks of misuse and population-level data would not be sufficient for the type of analysis suggested here

Response: Thank you for the comment. In order to define misuse, we need to have the diagnosis or indication that the patient was prescribed gabapentinoids for. On top of that, factors such as age, concomitant medication, comorbidities are needed for the analysis. Unfortunately, none of these are available in this dataset thus it would be hard to draw a reliable conclusion.

- Row 327: I completely take the authors' point that only licit sales are included in the data but not sure I fully follow the implications. My understanding is that there is no meaningful illegal manufacturing of gabapentinoids and diversion of gabapentinoids primarily occurs after a prescription has been issued/sale has been made. Assuming this is correct, this would mean that

even medications that would subsequently get diverted and sold in illicit markets would be captured by IQVIA data. In that case, from the perspective of this manuscript I don't think it matters whether the prescribed drugs were consumed by the original recipient or someone else or whether it was for medical reasons. In other words, the bigger question would be whether the prescription was consumed or not, rather than by whom and why. Or did I misunderstand and there is a concern that a notable share of gabapentinoid consumption is enabled by diversion from the healthcare system without involving any record (e.g., theft)? I suppose international trafficking of gabapentinoids could be another data issue, but again I am not aware that this would be a significant phenomenon?

Response: Thank you for the comment. We agree with the reviewer that illegal manufacturing of gabapentinoids could be unlikely. However, we could not completely rule out the possibility of off-record manufacturing of gabapentinoids that are not captured and happened before reaching the wholesaler, where MIDAS collects majority of its data. Therefore, we would like to keep this limitation in the discussion.

Lastly, I would suggest a few language edits:

- I understand that "abuse potential" is still language used by some stakeholders but in other instances, I would suggest avoiding the term "abuse." For instance rows 115-116, 274, 284, 326.

Response: Line 81-82 has been amended,

"The prominent use of gabapentinoids has raised concerns about their potential misuse..."

Line 195-196 has been amended,

"Another potential reason for the increase in gabapentinoid consumption is the increasing concerns over misuse of opioids and benzodiazepines^{34-36,37}."

Line 205-207 has been amended,

"Patients who have a history of opioid, benzodiazepine, or alcohol misuse have been reported as being vulnerable to misuse of gabapentinoids^{12,16,24,40}."

Line 246-248 have been amended,

"Individual patient data will be needed to study the appropriateness of gabapentinoid prescriptions or whether there was an increasing trend of misuse correlated to the growth in consumption."

- Row 186: would suggest changing to "annual percentage change of gabapentinoid consumption"

Line 108 has been amended,

"The average annual percentage change of gabapentinoid consumption was..."

- Row 278 and 341: "dependency": elsewhere in the manuscript and more often "dependence" is used

Response: We have made the changes accordingly.

"Clinicians, now more aware of the dependence and overdose issues with opioids and benzodiazepines, might have turned to prescribe gabapentinoids for the treatment of neuropathic pain, generalised anxiety, and insomnia^{4,38}." (page 10, line 199-202)

And,

"In view of the increasing concerns over their dependence and misuse potential, further studies are also needed..." (page 13, line 270-271)

- Row 308: would suggest changing "was potentially because of" to "may have been caused by"

We have made the change accordingly.

"The opposite trend of growth may have been caused by the decision of Pharmac, the pharmaceutical management agency in New Zealand⁴⁶, not to fund the use of pregabalin until December 2017⁴⁷." (page 11, line 230-232)

- Row 329: two sentences appear to have been mixed up

We apologise the confusion here. We changed the sentence as below. on

"Illicit sales of gabapentinoids are not captured and consequently the data presented in the study may not reflect the pattern of overall consumption in countries covered." (page 12, line 249-251)

Reviewer comments, second round

Reviewer #1 (Remarks to the Author):

Authors properly addressed most of the comments except for some of them as reported below:

- a) What was indication of use approved in different countries for each drug? What about reimbursement status and presence of generic? Was the cost of the drugs charged directly to the citizens in special conditions? Was there any major guideline potentially having impact on the drug use? Additional efforts should be put to retrieve information about those aspects which may substantially impact on the difference of gabapentinoids use across countries;
- b) The reasons why Puerto Rico is the country with the highest consumption of gabapentinoid in general in 2018 (Table 1), and gabapentin in particular (Figure S1), also need to be discussed

Authors provided an answer that should be concisely reported in the discussion

Reviewer #2 (Remarks to the Author):

I would like to thank the research team for the attention paid to responding my comments. The updated version addresses the points raised in my review. I had only two last comments for the authors' consideration but this is by no means something that needs to be actioned in the manuscript.

- I appreciated the additional details on sensitivity analyses and checks done by authors. I would suggest explicitly adding more information into the main text on these, if space allows. Specifically, I think it would be helpful to include details on the sensitivity analysis of imputed data and perhaps also a mention that a visual observation of the data did not give rise to any modelling concerns.
- Just to clarify my earlier comment, which now refers to line 246: I understand the desire to have individual-level data to assess trends in misuse, but my earlier point was to suggest that perhaps population-level survey data on rates of misuse would also be usable for a similar type of analysis. But this is definitely not a key point.

REVIEWER COMMENTS

Reviewer #1 (Remarks to the Author):

Authors properly addressed most of the comments except for some of them as reported below:

a) What was indication of use approved in different countries for each drug? What about reimbursement status and presence of generic? Was the cost of the drugs charged directly to the citizens in special conditions? Was there any major guideline potentially having impact on the drug use? Additional efforts should be put to retrieve information about those aspect which may substantially impact on the difference of gabapentinoids use across countries;

Response: We appreciate the comments from the reviewer. We would like to restate that the original purpose of this study is to describe the trend of gabapentinoid use in different countries but not attempting to explain rationales of such trend in different countries or globally. Other factors, such as presence of generics, cost of drugs/ reimbursement status, introduction of guidelines, coverage of healthcare, quality of healthcare, implementation of healthcare policies, etc. may have a complicated and multi-dimensional relationship with gabapentinoid consumption. Within country variation is also observed and the clinical practice of prescribers can be influenced by many other factors. Hence, a further study is warranted to investigate different potential predictors of the increase in gabapentinoid use as stated in line 286-288.

After reviewing the retrieved information as illustrated in the following paragraphs, we have included additional information into Supplementary information (Supplementary Table 1 ,3, 9-11). The following sentences have also been included in the discussion section.

“Other potential factors including but not limited to on and off-label indications, presence of generics, cost of drug, healthcare system, reimbursement status and relevant guidelines, (Supplementary Table 1, Supplementary Table 3 and Supplementary Table 9-11) may affect the drug utilisation pattern. To accurately assess the impact of these factors, a dedicated study utilising a single, comprehensive database platform is necessary. It is noteworthy that MIDAS, which focuses on national medication consumption data, does not provide sufficient information to thoroughly investigate the effects of these additional factors.” (Page 13, Line 271-278)

Regarding the approved indications in different countries, the following sentence has been added to the introduction section.

“The licensed indications of some of the countries/ regions are shown in Supplementary Table 1.” (Page 4, Line 72-73)

Regarding the presence of generics in different countries, based on the information from the MIDAS sales dataset, except for Japan, generics are present in all countries in 2018 for Gabapentin and Pregabalin. For Gabapentin enacarbil, neither the United States, Puerto Rico nor Japan have generic versions in the market. A remark has been included in Supplementary Table 3 to reflect this information.

Regarding the reimbursement status and cost of drugs, the classification of healthcare systems in all 65 countries/ regions and gabapentinoid reimbursement status in the selected 34 countries/ regions are listed in Supplementary Table 9 and 10.

The out-of-pocket payment for prescription medications, including gabapentinoids, varies even within the same country. Factors such as the state/ province that the patients resided at, the healthcare insurance plan that the patients were enrolled in, and the patients' demographics all affect the availability and out-of-pocket cost of gabapentinoids. For instance, a person who is above the age of 60 do not need to pay for their prescriptions in the UK but patients between the age of 18-60 do have to pay at a standard rate. In South Korea, the out-of-pocket payment may depend on the type of healthcare institute that a patient visited. In the United States, patients enrolled in Medicare and patients enrolled in Medicaid may pay for the same medication at a different rate, depending on the state that they are resided in or the indications of gabapentinoids that they were prescribed for. These variations happen in all 34 countries/ regions above. It will not be a fair and accurate approach to generalise the reimbursement status and out-of-pocket cost of gabapentinoids for all individual countries studied. From what we observed, gabapentinoids are included in the positive list of different insurance plans or healthcare systems of the 34 countries/ regions included above but some countries have limited their reimbursement status to certain conditions.

A dedicated study will need to be conducted instead to investigate the relationship between different healthcare reimbursement schemes with the consumption of prescription medications, including gabapentinoids.

Supplementary Table 11 shows the guidelines that are related to the use of gabapentinoids for a wide variety of medical conditions. Some guidelines are against the use of gabapentinoids while others recommend the use of them. The recommendations vary among different medical conditions. The above guidelines also cover other pharmacological agents that have similar indications as gabapentinoids, and the guidelines may not have a direct impact on the consumption of gabapentinoids.

The results of our study show a monotonic upward trend regarding gabapentinoid consumption across the span of 11 years. Some of the above guidelines may facilitate the consumption of gabapentinoids but they will not affect the conclusion of the study. Similar to the healthcare system or reimbursement status of medications in different countries. The relationship between the release of guidelines and consumption of gabapentinoids should be investigated in a separate study.

Supplementary Table 1. Licensed indications of use of Gabapentin, Pregabalin and Gabapentin Enacarbil (as of 31st May, 2023)

Country/ Region	Drug	Approved indication(s) of use/ Remarks
Canada	Gabapentin ¹	Adjunctive therapy for the management of patients with epilepsy who are not satisfactorily controlled by conventional therapy
	Pregabalin ¹	 a) Neuropathic pain associated with diabetic peripheral neuropathy b) Neuropathic pain associated with postherpetic neuralgia c) Neuropathic pain associated with spinal cord injury d) Pain associated with fibromyalgia
United States	Gabapentin ²	 a) Postherpetic neuralgia b) Adjunctive therapy in the treatment of partial onset seizures, with and without secondary generalization, in adults and paediatric patients 3 years and older with epilepsy
	Pregabalin ²	 a) Neuropathic pain associated with diabetic peripheral neuropathy b) Postherpetic neuralgia c) Adjunctive therapy for the treatment of partial-onset seizures in patients 17 years of age and older d) Fibromyalgia e) Neuropathic pain associated with spinal cord injury
	Gabapentin Enacarbil ²	 a) Moderate-to-severe primary restless legs syndrome in adults

		b) Postherpetic neuralgia in adults
Countries within European Medicine Agency (Austria, Belgium, Bulgaria, Croatia, Czech Republic, Estonia, Finland, France, Germany, Greece, Hungary, Ireland, Italy, Latvia, Lithuania, Luxembourg, Netherlands, Norway, Poland, Portugal, Romania, Slovakia, Slovenia, Spain and Sweden)	Gabapentin ³	 a) Adjunctive therapy in the treatment of partial seizures with and without secondary generalization in adults and children aged 6 years and above b) Monotherapy in the treatment of partial seizures with and without secondary generalization in adults and adolescents aged 12 years and above c) Peripheral neuropathic pain such as painful diabetic neuropathy and post-herpetic neuralgia in adults
	Pregabalin ⁴	 a) Peripheral and central neuropathic pain in adults b) Adjunctive therapy in adults with partial seizures with or without secondary generalisation c) Generalised anxiety disorder in adults
United Kingdom	Gabapentin ⁵	 a) Adjunctive treatment of focal seizures with or without secondary generalisation b) Monotherapy for focal seizures with or without secondary generalisation c) Peripheral neuropathic pain d) Menopausal symptoms, particularly hot flushes, in women with breast cancer e) Oscillopsia in multiple sclerosis f) Spasticity in multiple sclerosis g) Muscular symptoms in motor neurone disease

	Pregabalin ⁶	 a) Peripheral and central neuropathic pain b) Adjunctive therapy for focal seizures with or without secondary generalisation c) Generalised anxiety disorder
Australia	Gabapentin ⁷	 a) Partial seizures, including secondarily generalised tonic-clonic seizures, initially as add-on therapy in adults and children age 3 years and above who have not achieved adequate control with standard anti-epileptic drugs b) Neuropathic pain
	Pregabalin ⁷	 a) Neuropathic pain b) Adjunctive therapy in adults with partial seizures with or without secondary generalisation
New Zealand	Gabapentin ⁸	 a) Partial seizures, including secondarily generalised tonic-clonic seizures, initially as add-on therapy in adults and children age 3 years and above who have not achieved adequate control with standard anti-epileptic drugs b) Neuropathic pain
	Pregabalin ⁸	 a) Neuropathic pain b) Adjunctive therapy in adults with partial seizures with or without secondary generalisation
China	Gabapentin ⁹	 a) Postherpetic neuralgia b) Epilepsy: As an adjuvant treatment of partial seizures with or without secondary generalized seizures in adults and children over 12 years old. It can also be used as an adjuvant therapy for partial seizures in children aged 3-12

	Pregabalin ¹⁰	 a) Postherpetic neuralgia b) Fibromyalgia
Taiwan	Gabapentin ¹¹	 a) Adjuvant therapy for partial seizures in adults and children aged 3 or above b) Postherpetic neuralgia neuropathic pain
	Pregabalin ¹¹	 a) Neuropathic pain caused by diabetic peripheral neuropathy b) Postherpetic neuralgia c) Adjuvant treatment of partial epilepsy in adults d) Fibromyalgia e) Neuropathic pain caused by spinal cord injury
Japan	Gabapentin ¹²	 a) Treatment with concomitant antiepileptic drugs for partial seizures (including secondary generalized seizure) in patients with epilepsy for whom other antiepileptic drugs are not sufficiently effective b) Moderate to severe restless legs syndrome
	Pregabalin ¹³	 a) Neuropathic pain b) Pain associated with fibromyalgia
	Gabapentin Enacarbil ¹²	 a) Treatment with concomitant antiepileptic drugs for partial seizures (including secondary generalised seizure) in patients with epilepsy for whom other antiepileptic drugs are not sufficiently effective

		b) Moderate to severe restless legs syndrome
South Korea	Gabapentin ¹⁴	a) Epilepsy b) Neuropathic pain
	Pregabalin ¹⁴	a) Diabetic peripheral neuropathic pain b) Postherpetic neuralgia c) Neuropathic pain due to spinal cord injury d) Complex regional pain Syndrome e) Cancer neuropathic pain f) Post spinal surgery syndrome
Chile	Gabapentin ¹⁵	a) Monotherapy for the treatment of partial seizures with and without secondary generalization in adults and adolescents 12 years of age and older b) Adjunctive therapy in the treatment of partial seizures with and without secondary generalization in adults and children 3 years of age and older c) Diabetic neuropathic pain in adults and adolescents 12 years of age and older d) Post-herpetic neuralgia in adults and adolescents 12 years of age and older.
	Pregabalin ¹⁵	a) Neuropathic pain in adults

		 b) Adjunctive therapy in adults with partial seizures with or without secondary generalization c) Generalized anxiety disorder d) Fibromyalgia
Puerto Rico	Not found ¹⁶	Source: Portal del Departamento de Salud
Uruguay	Not found ¹⁷	Source: Ministry of Public Health Online drug information
Switzerland	Gabapentin ¹⁸	 a) Monotherapy in patients 12 years and older with partial-onset seizures with and without secondary generalization b) Adjunctive therapy in patients 3 years and older with partial seizures with and without secondary generalization c) Neuropathic pain associated with diabetic neuropathy in adults d) Neuropathic pain associated with post-herpetic neuralgia adults
	Pregabalin ¹⁸	 a) Peripheral and central neuropathic pain in adults b) Add-on therapy for partial onset seizures with or without secondary generalization in adult patients who have had an inadequate response to other antiepileptic drugs c) Generalized anxiety disorder
Kuwait	Not found ¹⁹	Source: Ministry of health
Saudi Arabia	Not found ²⁰	Source: Saudi Food and Drug Authority Drugs List

United Arab Emirates	Not found ²¹	Source: Ministry of Health and Prevention Registered Medical Product Directory
Argentina	Gabapentin ²²	 a) Adjunctive therapy in the treatment of epileptic patients in adults and children older than 6 years or more b) Monotherapy in the treatment of partial epilepsy attacks with and without secondary generalizations in adults and adolescents 12 years of age or older c) Diabetic neuropathic pain in adults d) Post-herpetic neuralgia in adults
	Pregabalin ²²	 a) Peripheral and central neuropathic pain in adults b) Adjunctive therapy in adults with partial seizures with or without secondary generalization c) Generalized anxiety disorder d) Fibromyalgia
Brazil	Not found ²³	Source: ANVISA - National Health Surveillance Agency Consultas
Colombia	Not found ²⁴	Source: Instituto Nacional de Vigilancia de Medicamentos y Alimentos
Ecuador	Not found ²⁵	Source: Agencia Nacional de Regulación, Control y Vigilancia Sanitaria Medication Consultation
Mexico	Not found ²⁶	Source: COFEPRIS Listados de Registros Sanitarios de Medicamentos
Peru	Not found ²⁷	Source: DIGEMID Consulta de Registro Sanitario de Productos Farmacéuticos

Venezuela	Not found	Not found (Drug regulatory authority not found)
Belarus	Not found ²⁸	Source: State Register of Medicinal Products of the Republic of Belarus
Russia	Not found ²⁹	Source: Ministry of Health of Russian Federation
Bosnia and Herzegovina	Not found ³⁰	Source: Agency for Medicinal Products and Medical Devices of Bosnia and Herzegovina
Serbia	Gabapentin ³¹	 a) Additional therapy for treatment of epilepsy b) Monotherapy for the treatment of epilepsy in adults and adolescents aged 12 and older c) Peripheral neuropathic pain, such as diabetic neuropathic pain or post-herpetic neuralgia
	Pregabalin ³¹	 a) Peripheral and central neuropathic pain b) As a supplement to existing therapy for partial convulsions with or without secondary generalizations in adults if the condition is not under control with existing therapy as a supplement c) Generalized anxiety disorder
Kazakhstan	Gabapentin ³²	Monotherapy or adjunctive therapy of partial epileptic seizures with or without secondary spread
	Pregabalin ³²	 a) Peripheral and central neuropathic pain b) Generalized anxiety disorder c) Adjunctive therapy in epilepsy with partial seizures with or without secondary generalizations

Jordan	Gabapentin ³³	 a) Adjunctive therapy in the treatment of partial seizures with and without secondary generalization in adults and children aged 6 years and above b) Monotherapy in the treatment of partial seizures with and without secondary generalization in adults and adolescents aged 12 years and above c) Peripheral neuropathic pain such as diabetic neuropathy and post-herpetic neuralgia
	Pregabalin ³³	 a) Peripheral and central neuropathic pain b) Adjunctive therapy in adults with partial seizures with or without secondary generalisation c) Fibromyalgia d) Generalised anxiety disorder
Lebanon	Not found ³⁴	Source: Lebanon National Drugs Database
Türkiye	Not found ³⁵	Source: Turkish Medicines and Medical Devices Agency
Thailand	Not found ³⁶	Source: Food and Drug Administration of Thailand
Algeria	Not found ³⁷	Source: Ministry of Health, Population and Hospital Reform of Algeria
South Africa	Not found ³⁸	Source: South African Health Products Regulatory Authority (SAHPRA)
Philippines	Not found ³⁹	Source: Food and Drug Administration of the Philippines
India	Not found ⁴⁰	Source: Central Drugs Standard Control Organization
Pakistan	Not found ⁴¹	Source: Drug Regulatory Authority of Pakistan. Registered drug index

Egypt	Not found ⁴²	Source: Egyptian Drug Authority
Morocco	Not found ⁴³	Source: Ministère de la Santé
Tunisia	Gabapentin ⁴⁴	a) Monotherapy treatment (including first-line treatment) in adults and children over 12 years of age, with simple and complex partial epileptic seizures, with or without secondary generalisation b) Adjunctive therapy in adults and children aged 3 years and over, presenting with simple and complex partial epileptic seizures, with or without secondary generalisation c) Neuropathic pain
	Pregabalin ⁴⁴	a) Neuropathic pain b) Adjunctive therapy for partial epileptic seizures with or without secondary generalisation c) Generalised anxiety disorder

Note: Licensed indications of Gabapentin, Pregabalin and Gabapentin enacarbil are identified in 41 countries/ regions as shown above. The above 41 countries/ regions have accounted for over 80% of the total consumption of gabapentinoids in 2018. The information above were retrieved following the hierarchy below:

- (1) National drug authority or National Formulary
- (2) Approved patient information leaflet or prescribing information uploaded online by the marketing authorisation holder (if licensed indications cannot be found from the first level of the hierarchy)
- (3) Reputable third party website that documents licensed indication(s) in the country/ region in interest (if licensed indications cannot be found from the first and second level of the hierarchy)

“Not found” will be shown under the “Drug” column if none of the above sources can provide information regarding approved use of gabapentinoids.

Presence of generics in all 65 countries/ regions (as of December 2018)

Country/ Region	Geographical Region	Gabapentin	Gabapentin enacarbil	Pregabalin
Canada	Northern America	Yes	NA	Yes
United States	Northern America	Yes	No	Yes
Chile	Central and South America and the Caribbean	Yes	NA	Yes
Puerto Rico	Central and South America and the Caribbean	Yes	No	Yes
Uruguay	Central and South America and the Caribbean	Yes	NA	Yes
Austria	Western Europe	Yes	NA	Yes
Belgium	Western Europe	Yes	NA	Yes
France	Western Europe	Yes	NA	Yes
Germany	Western Europe	Yes	NA	Yes
Luxembourg	Western Europe	Yes	NA	Yes
Netherlands	Western Europe	Yes	NA	Yes
Switzerland	Western Europe	Yes	NA	Yes
Estonia	Northern Europe	Yes	NA	Yes
Finland	Northern Europe	Yes	NA	Yes
Ireland	Northern Europe	Yes	NA	Yes
Latvia	Northern Europe	Yes	NA	Yes
Lithuania	Northern Europe	Yes	NA	Yes
Norway	Northern Europe	Yes	NA	Yes
Sweden	Northern Europe	Yes	NA	Yes
United Kingdom	Northern Europe	Yes	NA	Yes
Czech Republic	Eastern Europe	Yes	NA	Yes
Hungary	Eastern Europe	Yes	NA	Yes
Poland	Eastern Europe	Yes	NA	Yes
Slovakia	Eastern Europe	Yes	NA	Yes
Croatia	Southern Europe	Yes	NA	Yes
Greece	Southern Europe	Yes	NA	Yes
Italy	Southern Europe	Yes	NA	Yes

Portugal	Southern Europe	Yes	NA	Yes
Slovenia	Southern Europe	Yes	NA	Yes
Spain	Southern Europe	Yes	NA	Yes
Australia	Australia and New Zealand	Yes	NA	Yes
New Zealand	Australia and New Zealand	Yes	NA	Yes
Japan	Eastern Asia	No	No	No
South Korea	Eastern Asia	Yes	NA	Yes
Taiwan	Eastern Asia	Yes	NA	Yes
Kuwait	Western Asia	Yes	NA	Yes
Saudi Arabia	Western Asia	Yes	NA	Yes
United Arab Emirates	Western Asia	Yes	NA	Yes
Argentina	Central and South America and the Caribbean	Yes	NA	Yes
Brazil	Central and South America and the Caribbean	Yes	NA	Yes
Colombia	Central and South America and the Caribbean	Yes	NA	Yes
Ecuador	Central and South America and the Caribbean	Yes	NA	Yes
Mexico	Central and South America and the Caribbean	Yes	NA	Yes
Peru	Central and South America and the Caribbean	Yes	NA	Yes
Venezuela	Central and South America and the Caribbean	Yes	NA	Yes
Belarus	Eastern Europe	Yes	NA	Yes
Bulgaria	Eastern Europe	Yes	NA	Yes
Romania	Eastern Europe	Yes	NA	Yes
Russia	Eastern Europe	Yes	NA	Yes
Bosnia and Herzegovina	Southern Europe	Yes	NA	Yes
Serbia	Southern Europe	Yes	NA	Yes
China	Eastern Asia	Yes	NA	Yes
Kazakhstan	Central Asia	Yes	NA	Yes
Jordan	Western Asia	Yes	NA	Yes
Lebanon	Western Asia	Yes	NA	Yes
Türkiye	Western Asia	Yes	NA	Yes

Thailand	South-eastern Asia	Yes	NA	Yes
Algeria	Northern Africa	Yes	NA	Yes
South Africa	Southern Africa	Yes	NA	Yes
Philippines	South-eastern Asia	Yes	NA	Yes
India	Southern Asia	Yes	NA	Yes
Pakistan	Southern Asia	Yes	NA	Yes
Egypt	Northern Africa	Yes	NA	Yes
Morocco	Northern Africa	Yes	NA	Yes
Tunisia	Northern Africa	Yes	NA	Yes

Supplementary Table 9. Healthcare systems classification of the 65 countries/ regions

Country/ Region	Universal government-funded health system ^a	Universal public insurance system ^b	Universal public-private insurance ^c	Universal private health insurance system ^d	Non-universal insurance system ^e
Canada ⁴⁵	✓				
United States ⁴⁶					✓
Chile ⁴⁷			✓		
Puerto Rico ⁴⁸					✓
Uruguay ⁴⁹		✓			
Austria ⁵⁰		✓			
Belgium ⁵¹		✓			
France ⁵²		✓			
Germany ⁵³			✓		
Luxembourg ⁵⁴		✓			
Netherlands ⁵⁵				✓	
Switzerland ⁵⁶				✓	
Estonia ⁵⁷		✓			
Finland ⁵⁸	✓				
Ireland ⁵⁹	✓				
Latvia ⁶⁰		✓			
Lithuania ⁶¹		✓			
Norway ⁶²	✓				
Sweden ⁶³	✓				
United Kingdom ⁶⁴	✓				
Czech Republic ⁶⁵		✓			
Hungary ⁶⁶		✓			
Poland ⁶⁷		✓			
Slovakia ⁶⁸		✓			
Croatia ⁶⁹		✓			
Greece ⁷⁰		✓			

Italy ⁷¹	✓				
Portugal ⁷²	✓				
Slovenia ⁷³		✓			
Spain ⁷⁴	✓				
Australia ⁷⁵	✓				
New Zealand ⁷⁶	✓				
Japan ⁷⁷		✓			
South Korea ⁷⁸		✓			
Taiwan ⁷⁹		✓			
Kuwait ⁸⁰	✓				
Saudi Arabia ⁸¹	✓				
United Arab Emirates ⁸²				✓	
Argentina ⁸³			✓		
Brazil ⁸⁴	✓				
Colombia ⁸⁵		✓			
Ecuador ⁸⁶	✓				
Mexico ⁸⁷			✓		
Peru ⁸⁸			✓		
Venezuela ⁸⁹	✓				
Belarus ⁹⁰	✓				
Bulgaria ⁹¹		✓			
Romania ⁹²		✓			
Russia ⁹³		✓			
Bosnia and Herzegovina ⁹⁴		✓			
Serbia ⁹⁵		✓			
China ⁹⁶		✓			
Kazakhstan ⁹⁷	✓				
Jordan ⁹⁸					✓
Lebanon ⁹⁹					✓

Türkiye ¹⁰⁰			✓		
Thailand ¹⁰¹		✓			
Algeria ¹⁰²	✓				
South Africa ¹⁰³	✓				
Philippines ¹⁰⁴		✓			
India ¹⁰⁵	✓				
Pakistan ¹⁰⁶	✓				
Egypt ¹⁰⁷					✓
Morocco ¹⁰⁸					✓
Tunisia ¹⁰⁹					✓

^aUniversal government-funded health system: The universal healthcare services in these countries are covered by taxation and healthcare services are available to all citizens regardless of their income or employment status.

^bUniversal public insurance system: Work force in these countries are asked to join some form of social insurance programmes. These programmes are sometimes also funded by the employers and the government. People who are unemployed or belongs to special patient groups may have their healthcare expenses covered by the government or become ineligible for free healthcare.

^cUniversal public-private insurance: Countries included will ask their citizens to enrol in private insurance plans. For those who are ineligible, other plans from the government will cover their healthcare expenses.

^dUniversal private health insurance system: Private healthcare insurance is mandatory in these countries. Citizens from low-income families will have government subsidised private insurance schemes.

^eNon-universal insurance system: In these countries, not all citizens are insured. Some citizens may have private insurance coverage, while some others may have government subsidised public insurance plan. Patients who do not have insurance coverage will not be eligible for healthcare services.

Supplementary Table 10. Reimbursement status of gabapentinoids in selected countries/ regions (as of 31st May 2023)

	Reimbursement status of gabapentinoids
Canada ¹¹⁰⁻¹¹³	Under the Canada Health Act, prescription drugs administered in Canadian hospitals are provided to patients at cost. Outside of hospital settings, provincial and territorial governments are responsible for administering their own publicly funded drug plans. These public drug plans determine which prescription drugs are covered and under what conditions for eligible recipients. Many Canadians and their family members have drug coverage through their employment, while others may have no effective drug coverage and must pay the full cost of prescription medications. The rates of co-payments for prescription drugs also vary among provinces. For instance, in British Columbia, the Provincial and Territorial Public Drug Benefit Programme is called “Pharmacare”. It covers the cost of most drugs prescribed and the related dispensing fees with a maximum amount of CAD10. There are a total of 12 different plans under Pharamcare and they cover for different types of medications. However, eligibility of any of these plans will depend on the income level and medical conditions of the patient. One person can be covered by several plans. Generic versions of gabapentin and pregabalin, disregarding their prescribed quantity and indications, are included into the drug list of Pharamcare and neither of them require pre-approval. However, brand product of pregabalin is not funded under the scheme. The maximum amount that Pharmacare covers will also depend on the brand and strength prescribed. In Ontario, there are a total of 6 “Ontario Drug Benefit program” funded by the provincial government. The amount of deductible that needs to be paid by the patients will depend on their income and the type of programme they joined. Generic versions of gabapentin and pregabalin are included in the list of covered drugs of the programme.
United States ¹¹⁴⁻¹²¹	As of 2021, private health insurance is more commonly held than public coverage, with 66.0% of the population having private insurance and 35.7% having public coverage. Among the different types of health insurance, employer-based insurance is the most prevalent, covering 54.3% of the population, followed by Medicaid (18.9%), Medicare (18.4%), and direct-purchase coverage (10.2%). As different private insurance plans vary in drug reimbursement coverage, we will focus on the two major public healthcare insurance programmes in the United States, which are Medicare and Medicaid. Medicare is a federal health insurance program for individuals aged 65 or older. Individuals may also become eligible for Medicare if they have a disability, end-stage renal disease, or Lou Gehrig's disease. Part D of Medicare covers prescription drug reimbursement. Since 1st January, 2006, everyone with Medicare, regardless of income, health status, or prescription drug usage has had access to prescription drug coverage. Monthly premiums for Part D are required. Monthly premiums are required for Part D, and the premium amount may change annually. Additionally, patients may have to pay an additional amount each month based on their income. On the Medicare website, both gabapentin and pregabalin are included in the Part D drug plan formulary. The cost of the drugs paid by the patient will vary depending on the drug's "tier," the drug benefit phase the patient is in, the pharmacy used by the patient, and whether the patient receives "Extra Help" to cover Medicare drug coverage costs.

	Medicaid is a joint federal and state program that provides health coverage to individuals with limited income and resources. As Medicaid is administered by individual states, co-payments, coinsurance, deductibles, and similar charges may be imposed on most Medicaid-covered benefits, including inpatient and outpatient services. The specific amounts of these charges vary based on income. Each state also maintains its own preferred drug list, which determines the amount charged for each prescription. For drugs on the preferred list, the maximum allowable co-payment is USD4 per item for all patients, regardless of income. For non-preferred drugs, patients with an income at or below 150% of the Federal Poverty Level have a maximum allowable co-payment of USD8, while those with higher income levels pay 20% of the agency's cost. For instance, in New York state, the generic version of gabapentin and pregabalin are both listed as preferred drugs as anticonvulsants. Prior authorisation is required when they are used at a certain dose or concomitantly with opioids. Pregabalin solution and brand product of pregabalin are, however, listed as non-preferred drugs. In contrast, the state of Mississippi has indicated a more expansive use for gabapentin and pregabalin as their generic versions are listed as preferred agents for fibromyalgia and neuropathic pain. Gabapentin is also listed as preferred anticonvulsant adjuvant. Certain brands are, however, listed as non-preferred agents.
Countries within European Medicine Agency¹²²⁻¹²⁴ (Austria, Belgium, Bulgaria, Croatia, Czech Republic, Estonia, Finland, France, Germany, Greece, Hungary, Ireland, Italy, Latvia, Lithuania, Luxembourg, Netherlands, Norway, Poland, Portugal, Romania, Slovakia, Slovenia, Spain and Sweden)	According to the “Medicines Reimbursement Policies in Europe” published by the World Health Organisation regional office for Europe in 2018, different European Union (EU) member states adopt various eligibility criteria for drug reimbursement coverage, and they depend on the medicine (product-specific) or the disease the medicine aims to treat (disease-specific). Reimbursement eligibility may also be linked to a specific population group in need of medicines (population groups-specific) or the total medicine expenditure of a patient within a certain period of time (consumption-based). Product-specific reimbursement eligibility is determined based on factors such as therapeutic benefit, added value compared to alternatives, cost-effectiveness, and budget impact. Countries like the Netherlands, Italy, Czech Republic, and Slovakia primarily adopt this scheme. Disease-specific reimbursement eligibility involves a list of specific diseases for which pharmaceutical treatment is reimbursed. The reimbursement rates for the same medicine may differ depending on the patient's disease. Estonia, Latvia, Lithuania, Ireland, France, Bulgaria, and Portugal use this scheme either as the main or supplementary approach. Population groups-specific eligibility focuses on providing pharmaceutical reimbursement to specific groups in need, such as individuals requiring financial assistance due to their condition. Countries like Ireland, Finland, Hungary, Poland, and Romania combine this approach with other schemes. Consumption-based reimbursement eligibility is based on the patient's pharmaceutical consumption within a specified period. Once a defined threshold of out-of-pocket payments is reached, the public payer covers additional pharmaceutical expenses during that period. Sweden primarily follows this scheme. All included EU countries have reimbursement lists, with most applying a formulary indicating eligible drugs. Disease-specific reimbursement schemes often use a list of reimbursable diseases as the basis for coverage. Inclusion in an outpatient positive list doesn't guarantee full cost coverage; medicines may be partially reimbursed up to a specific percentage rate.

	Austria, Croatia, Germany, Ireland, Italy, and the Netherlands provide 100% reimbursement for publicly subsidized medicines, but other payments like prescription charges, deductibles, or reference price system fees may still apply. Most EU countries have a reference price system, where interchangeable medicines are grouped based on active substances or related subgroups. The public authority sets a reimbursed price for all medicines in the group. If the retail price exceeds the reference price, patients must pay the difference along with any applicable co-payments. The specifics of the reference price system vary among EMA countries. We selected two EU member states as examples to investigate whether gabapentin and pregabalin are include in their drug reference list. In Ireland, both gabapentin and pregabalin are included in the Health Service Executive Primary Care Reimbursement Service. In Germany, both medications are listed in the reference pricing lists of statutory health insurance funds. In Germany, both gabapentin and pregabalin are included in the reference pricing lists of the statutory health insurance fund.
United Kingdom ¹²⁵⁻¹²⁹	In the UK, the funding for individual drugs is determined by the local Integrated Care Boards (ICBs), formerly known as Clinical Commissioning Groups (CCGs). For example, both Gabapentin and Pregabalin are included in the North West London ICB's list of approved drugs. In 2023, the prescription charge in England is £9.65 per item. Patients have the option to purchase an NHS Prescription Prepayment Certificate, which costs £31.25 for a three-month period or £111.60 for a 12-month period. However, certain patient groups, such as those under 16, those aged 60 or over, and patients with specific medical conditions, are exempt from prescription charges and receive their medications free of charge. Prescriptions dispensed in Scotland under the NHS are free of charge. In NHS Wales, prescriptions are also provided free of charge. All prescriptions dispensed in Northern Ireland are free.
Australia ^{130,131}	The Australian federal government supports inpatient and outpatient care through the Medicare Benefits Scheme (MBS) and helps with outpatient prescription medicines through the Pharmaceutical Benefits Scheme (PBS). The Department of Health is responsible for overseeing national policies and programs related to healthcare, including the MBS and PBS. Medicare is the public health insurance system, and it is funded through national taxes, including a government levy. Private health insurance is widely accessible and offers coverage for out-of-pocket expenses and services from private healthcare providers. The government encourages individuals to enrol in private health insurance through tax rebates. The federal government subsidizes pharmaceuticals used in hospitals through the PBS. The cost-sharing for outpatient care may vary. Starting from 1st January 2023, patients may be required to pay up to AUD30.00 for most PBS medicines or AUD7.30 if they hold a concession card. The remaining cost, excluding brand premiums and certain charges, is covered by the Australian Government. From 1st January 2023, the PBS Safety Net threshold is set at AUD262.80 for patients with a concession card and AUD1,563.50 for other eligible patients. Once the Safety Net threshold is reached, general patients pay for further PBS prescriptions at the concessional co-payment rate, while concession card holders receive PBS prescriptions at no additional charge for the rest of the calendar year.

	Some medicines may have a price premium or brand premium, which is an additional payment made by the patient to the supplier for a specified brand of a PBS medicine. If a patient opts for a more expensive brand, they are responsible for paying the price difference to the drug company, in addition to the co-payment. The Pharmaceutical Benefits Scheme (PBS) maintains a schedule listing medicines available at government-subsidised prices. It is accessible to all Australian residents with a valid Medicare card. Both gabapentin and pregabalin are included under the PBS schedule list.
New Zealand ^{132,133}	All permanent residents in New Zealand have access to a comprehensive range of services that are primarily funded through pooled general taxes collected at the national level. As stated in the manuscript, Pharmac is the body which decides which medications should be funded by the New Zealand government. It only decided to fund gabapentin throughout most of the studied period (2008-2018). Despite being a medication long registered in the market, it only decided to fund pregabalin in 2017. Starting from 1st July, 2023, patients will no longer be required to pay a standard NZD5 prescription charge for each prescription. Previously, a co-payment of NZD5 per prescription was applicable for the first 20 prescription items collected by a patient or their family within a year. However, prescriptions from specialists and non-publicly funded prescribers will incur a co-payment of NZD15.
China ¹³⁴⁻¹³⁶	According to the “2022 Drug Catalogue” issued by Ministry of Human Resources and Social Security, both gabapentin and pregabalin are included as Class B drugs in the formulary. The "Class B List" is formulated by the state, and each province, autonomous region, and municipality directly under the Central Government may make appropriate adjustments on the list for not more than 15% of the total number of the drugs, according to the local economic level, medical needs, and prescribing habits. Under the national medical insurance scheme, patients are required to contribute a certain percentage of the cost for Class B drugs. Furthermore, additional expenses may be incurred according to the provisions of the basic medical insurance. The specific proportion of self-payment is determined by the district and reported to the relevant Labour and Social Security administrative department at the provincial, autonomous region, or municipal level. The public insurance programs only reimburse patients up to a certain limit, beyond which individuals are responsible for covering all out-of-pocket costs. The ceiling for reimbursement varies based on factors such as the patient's occupation, place of residence, and other considerations.
Taiwan ^{79,137}	In Taiwan, the National Health Insurance (NHI) program is mandatory for almost the entire population, including all nationals who have had a registered domicile in Taiwan for six months or more and all new-borns. The NHI premiums are jointly paid by the insured individuals, group insurance applicants, and the government. The premium rates for NHI have undergone several adjustments over the years. From January 2016, the general premium rate was set at 4.69%, and supplementary premiums were introduced at a rate of 1.91%. The premiums payable by insured individuals depend on their category, and they are calculated based on the insured's premium rateable wage multiplied by the general premium rate.

	NHI provides coverage for a wide range of healthcare services, including outpatient care, inpatient care, traditional Chinese medicine, dental care, child delivery, rehabilitation, home health care, and chronic mental illness rehabilitation. Medical payments under NHI cover various aspects such as diagnosis, examinations, laboratory tests, surgeries, anaesthesia, medications, materials, treatments, nursing, and insured beds. In terms of pharmacy services, patients with a prescription from a contracted hospital or clinic can obtain medications at any contracted pharmacy. Co-payments are required for physician visits and prescription drugs, while coinsurance applies to inpatient care, subject to limits and exemptions. The co-payment for outpatient prescription drugs is capped at TWD200 per visit, regardless of the number of prescribed drugs. However, there is no annual cap on drug co-payments. As of May 2023, only pregabalin is included in the drug list under the National Health Insurance, and there are specific restrictions on the insured indications for which it can be prescribed.
Japan ¹³⁸⁻¹⁴¹	Japan's statutory health insurance system (SHIS) is funded primarily by taxes and individual contributions. Enrollment in either an employment-based or residence-based health insurance plan is mandatory for all citizens. Benefits include hospital, primary, specialty, and mental health care, as well as prescription drugs. The system covers 98.3% of the population, with the remaining 1.7 percent covered by the Public Social Assistance Program. Health expenditures are funded by taxes (42%), mandatory individual contributions (42%), and out-of-pocket charges (14%). Contributions are shared between employers and employees in employment-based plans, while the national government, prefectures, and municipalities contribute to residence-based plans. In employment-based health insurance plans, both employers and employees are required to contribute to the premiums. The contribution rates are typically around 10% of monthly salaries and bonuses and are based on the employee's income. There is a cap on the contribution amount. These contributions are tax-deductible and can vary depending on the type of insurance fund and the prefecture. For residence-based health insurance plans, the national government, along with prefectures and municipalities, provides funding for a portion of individuals' mandatory contributions. It is worth noting that while more than 70% of the population holds some form of voluntary private health insurance, these private plans primarily serve as supplementary or complementary coverage alongside the statutory health insurance system. All statutory health insurance plans in Japan offer the same benefits package, which includes hospital, primary, specialty, and mental health care, as well as prescription drugs. Enrolees typically pay a fixed percentage (usually 30%) as coinsurance or co-payment for health services and pharmaceuticals. However, the coinsurance rate may vary for certain special patient groups such as the elderly or children. The system also incorporates catastrophic coverage, with monthly out-of-pocket thresholds and an annual household out-of-pocket ceiling based on income and age. In Japan, medication can be dispensed by both clinics and pharmacies. While doctors can directly provide medication to patients, the use of pharmacists has been growing, with the majority of prescriptions being filled at pharmacies.

	The Central Social Insurance Medical Council plays a role in determining the SHIS list of covered pharmaceuticals and their prices. Price revisions for pharmaceuticals and medical devices are based on market surveys. The NHI drug price represents the final retail price of pharmaceuticals charged to patients and is set uniformly across the country. Both Gabapentin and Pregabalin are included in the latest SHIS list in May 2023.
South Korea ¹⁴²⁻¹⁴⁵	The healthcare system in South Korea consists of two main components: mandatory social health insurance and medical aid. The National Health Insurance (NHI) system ensures healthcare coverage for all citizens, funded by contributions from the insured individuals and government subsidies. The medical aid program, on the other hand, provides healthcare services to low-income groups through government subsidies. From 1st January, 2022, the NHI premium rate, which includes long-term care insurance, is approximately 7.85% of monthly wages, with a cap on monthly contributions. The premium is split equally between employers and employees. The Ministry of Health and Welfare (MoHW) is responsible for overseeing the National Health Insurance system, which includes two key institutions: the National Health Insurance Service (NHIS) and the Health Insurance Review & Assessment Service (HIRA). The NHIS serves as the insurer, while HIRA conducts reviews of claims and assesses the quality of healthcare services. HIRA plays a role in determining the eligibility of products for reimbursement under the NHI system. The price recommended by HIRA sets a maximum reimbursement price for the drug, and when a product becomes off-patent and generics are available, the reimbursement price for the brand product is reduced. The amount patients need to pay out-of-pocket, including medication fees, depends on the healthcare institute they visit. Visiting a tertiary general hospital usually results in higher out-of-pocket payments compared to clinics or other healthcare institutes. The out-of-pocket payment percentage may also vary for certain special patient groups. Various formulations of Gabapentin and Pregabalin are eligible for reimbursement under the National Health Insurance scheme, and their respective maximum reimbursement prices are listed on the HIRA website.

Supplementary Table 11. Published guidelines related to the use of gabapentinoids between 2008-2018

Country/ Region	Name of the guideline	Year	Issuing Body	Extracted context that is related to the use of gabapentinoids
United Kingdom	Generalised anxiety disorder and panic disorder in adults: management ¹⁴⁶	2011	National Institute for Health and Care Excellence (NICE)	- If the person cannot tolerate selective serotonin reuptake inhibitor (SSRIs) or serotonin–norepinephrine reuptake inhibitor (SNRIs), consider offering pregabalin.
	Epilepsies in children, young people and adults ¹⁴⁷	2012, update in 2022	NICE	 - Pregabalin is considered as a second-line add-on treatment option for focal seizures. - Gabapentin and pregabalin are included as medications that may exacerbate absence seizures. - Gabapentin and pregabalin are not recommended to use in myoclonic seizures as they may exacerbate seizures. - Gabapentin and pregabalin are included as medications that may exacerbate atonic seizures. - Gabapentin and pregabalin may exacerbate seizures in people with Dravet syndrome. - Gabapentin and pregabalin may exacerbate seizures in people with Lennox–Gastaut syndrome. - Gabapentin and pregabalin are included as medications that may exacerbate myoclonic-atonic seizures.
	Neuropathic pain in adults: pharmacological management in non-specialist settings ¹⁴⁸	2013	NICE	 - All neuropathic pain (except trigeminal neuralgia)  ➤ Offer a choice of amitriptyline, duloxetine, gabapentin or pregabalin as initial treatment for neuropathic pain (except trigeminal neuralgia). See additional information for more on duloxetine, gabapentin and pregabalin.
	Evidence-based pharmacological treatment of anxiety disorders, post-traumatic stress disorder and obsessive-compulsive disorder ¹⁴⁹	2014	British Association for Psychopharmacology (BAP)	- Pregabalin is recommended as one of the treatment options for acute treatment of generalised anxiety disorder.

				 - The dosage of pregabalin is also recommended to be increased if the initial treatments fail. - Pregabalin augmentation is recommended after a non-response to initial SSRI or SNRI treatment.
	Restless legs syndrome: Oxycodone/naloxone prolonged release ¹⁵⁰	2015	NICE	 - Oxycodone/naloxone should only be considered after failure of dopaminergic therapy. Other non-dopaminergic drug treatment options for restless legs syndrome include off-label use of pregabalin, gabapentin, clonazepam or weak opioids such as codeine. - First-line drug treatment options for people with frequent or daily symptoms include non-ergot dopamine agonists (for example, pramipexole, ropinirole or rotigotine); pregabalin (off-label use) and gabapentin (off-label use).
Europe	EFNS guidelines on the pharmacological treatment of neuropathic pain: 2010 revision ¹⁵¹	2010	European Federation of Neurological Societies	 - Both gabapentin and pregabalin are both listed as level A rating for treatment of neuropathic pain related to diabetic neuropathy and postherpetic neuralgia. - Pregabalin is listed as level A rating for treatment of central pain related neuropathic pain.
Canada	Canadian clinical practice guidelines for the management of anxiety, post-traumatic stress and obsessive-compulsive disorders ¹⁵²	2014	Anxiety Disorders Association of Canada (ADAC)	 - Gabapentin is included as third line treatment for panic disorder. - Pregabalin and Gabapentin are listed as first-line and second-line treatment option, respectively, for social anxiety disorder. - Pregabalin monotherapy is included as the first-line treatment option and as second-line adjunctive therapy option for generalised anxiety disorder. - Pregabalin is listed as third-line adjunctive therapy option for obsessive-compulsive disorder. - Pregabalin and Gabapentin are both listed as third-line adjunctive therapy option for posttraumatic stress disorder.

	Pharmacological management of chronic neuropathic pain: Revised consensus statement from the Canadian Pain Society ¹⁵³	2014	Canadian Pain Society	- Gabapentinoids (gabapentin and pregabalin) are recommended as first-line treatment options for neuropathic pain.
United States	Practice guideline for the treatment of Patients with Panic Disorder ¹⁵⁴	2009	American Psychiatric Association (APA)	- After first- and second-line treatments and augmentation strategies have been exhausted (either due to lack of efficacy or intolerance of the treatment by the patient), less well-supported treatment strategies may be considered. These include monotherapy or augmentation with gabapentin or a second-generation antipsychotic or with a psychotherapeutic intervention other than Cognitive behavioural therapy (CBT) or Panic-Focused Psychodynamic Psychotherapy (PfPP).
	Evidence-based guideline: Treatment of painful diabetic neuropathy ¹⁵⁵	2011	American Academy of Neurology (AAN)	- Pregabalin is established as effective and should be offered for relief of painful diabetic neuropathy (PDN) (Level A). - Gabapentin is probably effective and should be considered for treatment of PDN (Level B).
	Management of Postoperative Pain: A Clinical Practice Guideline ¹⁵⁶	2016	the American Pain Society; the American Society of Regional Anesthesia and Pain Medicine; the American Society of Anesthesiologists' Committee on Regional Anesthesia	- Gabapentin and pregabalin are included as medications that can be administered pre-operatively for Thoracotomy, Open laparotomy, Total hip replacement, Total knee replacement, Spinal fusion and Coronary artery bypass grafting.
	Practice guideline update summary: Efficacy and tolerability of the new antiepileptic drugs I: Treatment of new-onset epilepsy ¹⁵⁷	2018	American Academy of Neurology (AAN)	- Gabapentin may (Level C) be considered in decreasing seizure frequency in patients ≥ 60 years with new-onset focal epilepsy. - No high-quality studies suggest pregabalin is effective in treating new-onset epilepsy because no high-quality studies exist in adults of various ages.
	Practice guideline update summary: Efficacy and tolerability of the new antiepileptic drugs II: Treatment-resistant epilepsy ¹⁵⁸	2018	American Academy of Neurology (AAN)	- Immediate-release pregabalin (Level A) is established as effective to reduce seizure frequency for treatment-resistant adult focal epilepsy.
International	Guidelines for the pharmacological treatment of anxiety disorders,	2012	International Journal of Psychiatry in Clinical Practice	- Pregabalin is recommended as one of the first-line drugs for generalized anxiety disorder.

	obsessive–compulsive disorder and posttraumatic stress disorder in primary care ¹⁵⁹			
	Pharmacotherapy for neuropathic pain in adults: systematic review, meta-analysis and updated NeuPSIG recommendations ¹⁶⁰	2015	The Neuropathic Pain Special Interest Group (NeuPSIG)	- Both gabapentin and pregabalin are included as first-line options for treatment of neuropathic pain.

b) The reasons why Puerto Rico is the country with the highest consumption of gabapentinoid in general in 2018 (Table 1), and gabapentin in particular (Figure S1), also need to be discussed. Authors provided an answer that should be concisely reported in the discussion.

Response: We thank the reviewer for suggesting us to include the answer into the discussion. The suggested explanation has now been included as below.

“As shown in Table 1 and Figure 2, Puerto Rico and the US had the highest consumption rate of gabapentinoids in 2018. They also share a similar trend in gabapentinoid consumption across the years, with gabapentin taking up the majority of the increase in consumption. This phenomenon may be explained by the fact that Puerto Rico is a territory of the US. Many of Puerto Rico’s healthcare professionals are trained in the US and may share similar clinical beliefs as their US counterparts.”
(Page 11, Line 231-236)

Reviewer #2 (Remarks to the Author):

I would like to thank the research team for the attention paid to responding my comments. The updated version addresses the points raised in my review. I had only two last comments for the authors' consideration but this is by no means something that needs to be actioned in the manuscript.

- I appreciated the additional details on sensitivity analyses and checks done by authors. I would suggest explicitly adding more information into the main text on these, if space allows. Specifically, I think it would be helpful to include details on the sensitivity analysis of imputed data and perhaps also a mention that a visual observation of the data did not give rise to any modelling concerns.

Response: We would like to thank the reviewer for the constructive comment. As stated on page 16 Line 337-339, the imputation of missing product strength is based on the most-sold product of the same drug, in other words, the median strength of that product. Visual observation will not demonstrate a difference to the existing model. The following has been included into the manuscript:

"To investigate whether imputation of missing strength of the product affected the finding of the study, we performed another set of sensitivity analysis by removing products that have missing strength." (P.17, Line 357-360)

"The other sensitivity analysis that removed products with imputed strength has a pooled multinational consumption levels of 4.11 DDD/TID (95%CI, 2.99 to 5.66) in 2008 and 17.84 DDD/TID (95%CI, 13.39 to 23.77) in 2018, which are comparable to our main analysis." (P.7, Line 139-141)

- Just to clarify my earlier comment, which now refers to line 246: I understand the desire to have individual-level data to assess trends in misuse, but my earlier point was to suggest that perhaps population-level survey data on rates of misuse would also be usable for a similar type of analysis. But this is definitely not a key point.

Response: We thank the reviewer for the comment and suggesting us to find population-level survey data on rates of misuse regarding gabapentinoids. Unfortunately, such population-level data is neither included in the dataset nor accessible through other public sources for all countries/ regions included in our study.

We attempted to retrieve such information from Global Health Data Exchange website:
<https://ghdx.healthdata.org/>

However, it does not have specific misuse data for gabapentinoids.

References

- 1 Health Canada. *Drug Product Database online query*, <<https://health-products.canada.ca/dpd-bdpp/dispatch-repartition>> (2023).
- 2 U.S. Food and Drug Administration. *FDA Label Search*, <<https://labels.fda.gov/getIngredientName.cfm>> (2023).
- 3 European Medicines Agency. *Neurontin - Article 30 referral - Annex I, II, III*, <https://www.ema.europa.eu/en/documents/referral/neurontin-article-30-referral-annex-i-ii-iii_en.pdf> (2023).
- 4 European Medicines Agency. *14/03/2023 Lyrica - EMEA/H/C/000546 - N/0122*, <https://www.ema.europa.eu/en/documents/product-information/lyrica-epar-product-information_en.pdf> (2023).
- 5 National Institute for Health and Care Excellence. *Gabapentin*, <<https://bnf.nice.org.uk/drugs/gabapentin/>> (2023).
- 6 National Institute for Health and Care Excellence. *Pregabalin*, <<https://bnf.nice.org.uk/drugs/gabapentin/>> (2023).
- 7 Therapeutic Goods Administration. *Product and Consumer Medicine Information*, <<https://www.ebs.tga.gov.au/ebs/picmi/picmirepository.nsf/PICMI?OpenForm&t=&q=gabapentin>> (2023).
- 8 Medsafe New Zealand Medicines and Medical Devices Safety Authority. *Data Sheets and Consumer Medicine Information*, <<https://www.medsafe.govt.nz/Medicines/SearchResult.asp>> (2023).
- 9 药智网. 加巴喷丁, <<https://db.yaozh.com/pijian?name=%E5%8A%A0%E5%B7%B4%E5%96%B7%E4%B8%81>> (2023).
- 10 Pfizer Manufacturing Deutschland GmbH. *普瑞巴林胶囊说明书*, <<https://viatris.cn/-/media/project/viatris-sites/product-document/06pregabalin.pdf>> (2023).
- 11 Food and Drug Administration. *西藥、醫療器材、化粧品許可證查詢*, <<https://info.fda.gov.tw/>> (2022).
- 12 Pharmaceuticals and Medical Devices Agency. *Summary of investigation results Gabapentin, Gabapentinencarbil*, <<https://www.pmda.go.jp/files/000211782.pdf>> (2016).
- 13 Pharmaceuticals and Medical Devices Agency. *Pregabalin*, <<https://www.pmda.go.jp/files/000197715.pdf#page=13>> (2014).
- 14 Health Insurance Review & Assessment Service. *가 가 가 가 가 가*, <<https://www.hira.or.kr/rc/insu/insuadtcrt/InsuAdtCrtrList.do?pgmid=HIRAA030069000400>> (2022).
- 15 Instituto de Salud Pública de Chile. *Consulta Productos Registrados*, <<https://registrosanitario.ispch.gob.cl/>> (2023).
- 16 Departamento de Salud. *Portal del Departamento de Salud*, <<https://www.salud.gov.pr/>> (2023).
- 17 Ministerio de Salud Pública. *Consulta de Medicamentos*, <<https://www.gub.uy/ministerio-salud-publica/home>> (2023).
- 18 Swissmedic. *Medicinal product information*, <<https://www.swissmedic.ch/swissmedic/en/home/services/medicinal-product-information.html>> (2019).
- 19 health, M. o. *MOH Kuwait*, <<https://www.moh.gov.kw>> (2023).
- 20 Saudi Food and Drug Authority. *Drugs list*, <<https://www.sfda.gov.sa/en/drugs-list>> (2023).
- 21 Ministry of Health and Prevention. *Registered Medical Product Directory*, <<https://mohap.gov.ae/en/services/registered-medical-product-directory>> (2023).

- 22 National Administration of Drugs, F. a. M. D. *VADEMECUM NACIONAL DE MEDICAMENTOS*, <<http://anmatvademecum.servicios.pami.org.ar/index.html>> (2023).
- 23 ANVISA - AGÊNCIA NACIONAL DE VIGILÂNCIA SANITÁRIA. *Consultas* <<https://consultas.anvisa.gov.br/>> (2023).
- 24 Instituto Nacional de Vigilancia de Medicamentos y Alimentos. *Alertas sanitarias medicamentos y productos biológicos*, <<https://www.invima.gov.co>> (2023).
- 25 Agencia Nacional de Regulación Control y Vigilancia Sanitaria. *Medication Consultation*, <<https://www.controlsanitario.gob.ec/>> (2023).
- 26 COFEPRIS - Mexico Ministry of Health. *Listados de Registros Sanitarios de Medicamentos*, <<https://www.gob.mx/cofepris/documentos/registros-sanitarios-medicamentos>> (2023).
- 27 Dirección General de Medicamentos Insumos y Drogas (DIGEMID). *Consulta de Registro Sanitario de Productos Farmacéuticos*, <<https://www.digemid.minsa.gob.pe/productospesquisados/principal/consultaproductospesquisados.aspx>> (2023).
- 28 Center for Expertise and Testing in Healthcare. *State Register of Medicinal Products of the Republic of Belarus*, <<https://www.rceth.by/Refbank>> (2023).
- 29 Ministry of Health of Russian Federation. *Ministry of Health of the Russian Federation*, <<https://minzdrav.gov.ru/>> (2023).
- 30 Agency for Medicinal Products and Medical Devices of Bosnia and Herzegovina. *Medicinal Products*, <<http://www.almbih.gov.ba/en/medicinal-products/>> (2023).
- 31 Medicines and Medical Devices Agency of Serbia. *Search of medicines and medical devices*, <<https://www.alims.gov.rs/english/medicinal-products/search-for-human-medicines/>> (2023).
- 32 Ministry of Health of the Republic of Kazakhstan. *National drug formulary of Kazakhstan*, <<https://www.knf.kz/kk>> (2023).
- 33 Jordan Food and Drug Administration. *Drug information, prices and leaflets*, <<http://www.jfda.jo/>> (2023).
- 34 Ministry of Public Health. *Lebanon National Drugs Database*, <<https://www.moph.gov.lb/en/Pages/3/3010/pharmaceuticals#/en/Drugs/index/3/3974/lebanon-national-drugs-database>> (2023).
- 35 Turkish Medicines and Medical Devices Agency. *List of Licensed Medicinal Products for Human Use*, <<https://www.titck.gov.tr>> (2023).
- 36 Food and Drug Administration. *Searching system for pharmaceutical products*, <<https://porta.fda.moph.go.th/>> (2023).
- 37 Ministry of Health, P. a. H. R. *Ministry of Health, Population and Hospital Reform*, <<http://www.sante.gov.dz>> (2023).
- 38 South African Health Products Regulatory Authority (SAHPRA) *Registered health products*, <<https://www.sahpra.org.za/databases-registers/>> (2023).
- 39 Food and Drug Administration. <<https://www.fda.gov.ph/>> (2023).
- 40 Central Drugs Standard Control Organization. *Drugs*, <<https://cdsco.gov.in/>> (2023).
- 41 Drug Regulatory Authority of Pakistan. *Registered drug index*, <<https://www.dra.gov.pk>> (2023).
- 42 Egyptian Drug Authority. *Egyptian Pharmacopoeia (content under construction)*, <<https://edaegypt.gov.eg/en/>> (2023).
- 43 Ministère de la Santé. *Ministère de la Santé*, <<https://www.sante.gov.ma/Pages/Accueil.aspx>> (2023).
- 44 Direction de la Pharmacie et du Médicament. *la liste des Médicaments ayant l'A.M.M. ,* <<http://www.dpm.tn/>> (2023).
- 45 Government of Canada. *Canada's Health Care System*, <<https://www.canada.ca/en/health-canada/services/health-care-system/reports-publications/health-care-system/canada.html>> (2019).

- 46 Roosa Tikkanen, R. O., Elias Mossialos, Ana Djordjevic, George A. Wharton. *International Health Care System Profiles - United States*,
<<https://www.commonwealthfund.org/international-health-policy-center/countries/United-States>> (2020).
- 47 Bastias, G., Pantoja, T., Leisewitz, T. & Zarate, V. Health care reform in Chile. *CMAJ* **179**, 1289-1292 (2008). <https://doi.org/10.1503/cmaj.071843>
- 48 The Office of the Assistant Secretary for Planning and Evaluation (ASPE). (ed USA Department of Health & Human Services) (2017).
- 49 Pan American Health Organisation. Functioning of the health system in Uruguay - Principles, financing, management, and care model. (Pan American Health Organisation and Regional Office for the Americas - World Health Organization, 2020).
- 50 Bachner, F. *et al.* Austria: Health System Review. *Health Syst Transit* **20**, 1-254 (2018).
- 51 Gerkens, S. & Merkur, S. Belgium: Health System Review. *Health Syst Transit* **22**, 1-237 (2020).
- 52 Chevreul, K., Berg Brigham, K., Durand-Zaleski, I. & Hernandez-Quevedo, C. France: Health System Review. *Health Syst Transit* **17**, 1-218, xvii (2015).
- 53 Blumel, M., Spranger, A., Achstetter, K., Maresso, A. & Busse, R. Germany: Health System Review. *Health Syst Transit* **22**, 1-272 (2020).
- 54 OECD, Systems, E. O. o. H. & Policies. *Luxembourg: Country Health Profile 2021*. (2021).
- 55 Kroneman, M. *et al.* Netherlands: Health System Review. *Health Syst Transit* **18**, 1-240 (2016).
- 56 De Pietro, C. *et al.* Switzerland: Health System Review. *Health Syst Transit* **17**, 1-288, xix (2015).
- 57 Habicht, T. *et al.* Estonia: Health System Review. *Health Syst Transit* **20**, 1-189 (2018).
- 58 Keskimaki, I. *et al.* Finland: Health System Review. *Health Syst Transit* **21**, 1-166 (2019).
- 59 McDaid, D., Wiley, Miriam, Maresso, Anna. *et al.* Ireland: health system review. (2009).
- 60 Behmane, D. *et al.* Latvia: Health System Review. *Health Syst Transit* **21**, 1-165 (2019).
- 61 Murauskiene, L., Janoniene, R., Veniute, M., van Ginneken, E. & Karanikolos, M. Lithuania: health system review. *Health Syst Transit* **15**, 1-150 (2013).
- 62 Saunes, I. S., Karanikolos, M. & Sagan, A. Norway: Health System Review. *Health Syst Transit* **22**, 1-163 (2020).
- 63 Anell, A., Glenngard, A. H. & Merkur, S. Sweden health system review. *Health Syst Transit* **14**, 1-159 (2012).
- 64 Anderson, M. *et al.* United Kingdom: Health System Review. *Health Syst Transit* **24**, 1-194 (2022).
- 65 Bryndova, L. *et al.* Czechia: Health System Review. *Health Syst Transit* **25**, 1-216 (2023).
- 66 Gaal, P., Szigeti, S., Csere, M., Gaskins, M. & Panteli, D. Hungary health system review. *Health Syst Transit* **13**, 1-266 (2011).
- 67 Sowada, C. *et al.* Poland: Health System Review. *Health Syst Transit* **21**, 1-234 (2019).
- 68 Smatana, M. *et al.* Slovakia: Health System Review. *Health Syst Transit* **18**, 1-210 (2016).
- 69 Dzakula, A. *et al.* Croatia: Health System Review. *Health Syst Transit* **23**, 1-146 (2021).
- 70 Economou, C., Kaitelidou, D., Karanikolos, M. & Maresso, A. Greece: Health System Review. *Health Syst Transit* **19**, 1-166 (2017).
- 71 Giulio de Belvis, A. *et al.* Italy: Health System Review. *Health Syst Transit* **24**, 1-236 (2022).
- 72 de Almeida Simoes, J., Augusto, G. F., Fronteira, I. & Hernandez-Quevedo, C. Portugal: Health System Review. *Health Syst Transit* **19**, 1-184 (2017).
- 73 Albrecht, T. *et al.* Slovenia: Health System Review. *Health Syst Transit* **23**, 1-183 (2021).
- 74 Bernal-Delgado, E. *et al.* Spain: Health System Review. *Health Syst Transit* **20**, 1-179 (2018).
- 75 Australian Government - Department of Health and Aged Care. *The Australian health system*, <<https://www.health.gov.au/about-us/the-australian-health-system>> (2019).

- 76 Jacqueline Cumming, J. M., Colin Barr, Greg Martin, Zac Gerring, Jacob Daubé. New Zealand health system review. *Health systems in transition* **Vol. 4** (2014).
- 77 Haruka Sakamoto, M. R., Shuhei Nomura, Etsuji Okamoto, Soichi Koike, Hideo Yasunaga, Norito Kawakami, Hideki Hashimoto, Naoki Kondo, Sarah Krull Abe, Matthew Palmer, Cyrus Ghaznavi. Japan health system review. *Health Systems in Transition* **Vol. 8** (2018).
- 78 Soonman Kwon, T.-j. L., Chang-yup Kim. Republic of Korea health system review. *Health systems in transition* **Vol. 5** (2015).
- 79 Cheng, T.-M. *International Health Care System Profiles - Taiwan*, <<https://www.commonwealthfund.org/international-health-policy-center/countries/taiwan>> (2020).
- 80 World Health Organization. Country Cooperation Strategy at a glance - Kuwait. (World Health Organization, 2017).
- 81 Alkhamis, A. Health care system in Saudi Arabia: an overview. *East Mediterr Health J* **18**, 1078-1079 (2012). <https://doi.org/10.26719/2012.18.10.1078>
- 82 Koornneef, E., Robben, P. & Blair, I. Progress and outcomes of health systems reform in the United Arab Emirates: a systematic review. *BMC Health Serv Res* **17**, 672 (2017). <https://doi.org/10.1186/s12913-017-2597-1>
- 83 Bello, M. & Becerril-Montekio, V. M. [The health system of Argentina]. *Salud Publica Mex* **53 Suppl 2**, s96-s108 (2011).
- 84 OECD. *OECD Reviews of Health Systems: Brazil 2021*. (2021).
- 85 OECD. *OECD Reviews of Health Systems: Colombia 2016*. (2015).
- 86 Flores Jimenez, S. E. & San Sebastian, M. Assessing the impact of the 2008 health reform in Ecuador on the performance of primary health care services: an interrupted time series analysis. *Int J Equity Health* **20**, 169 (2021). <https://doi.org/10.1186/s12939-021-01495-2>
- 87 Gonzalez Block, M. A., Reyes Morales, H., Hurtado, L. C., Balandran, A. & Mendez, E. Mexico: Health System Review. *Health Syst Transit* **22**, 1-222 (2020).
- 88 OECD. *OECD Reviews of Health Systems: Peru 2017*. (2017).
- 89 Page, K. R. *et al.* Venezuela's public health crisis: a regional emergency. *Lancet* **393**, 1254-1260 (2019). [https://doi.org/10.1016/S0140-6736\(19\)30344-7](https://doi.org/10.1016/S0140-6736(19)30344-7)
- 90 Richardson, E., Malakhova, I., Novik, I. & Famenka, A. Belarus: health system review. *Health Syst Transit* **15**, 1-118 (2013).
- 91 Dimova, A. *et al.* Bulgaria: Health System Review. *Health Syst Transit* **20**, 1-230 (2018).
- 92 Vladescu, C., Scintee, S. G., Olsavszky, V., Hernandez-Quevedo, C. & Sagan, A. Romania: Health System Review. *Health Syst Transit* **18**, 1-170 (2016).
- 93 Popovich, L. *et al.* Russian Federation. Health system review. *Health Syst Transit* **13**, 1-190, xiii-xiv (2011).
- 94 Juliane Winkelmann, Y. L., Boris Rebac. Health Systems in Action: Bosnia and Herzegovina. 28 (World Health Organization. Regional Office for Europe., 2022).
- 95 Bjegovic-Mikanovic, V. *et al.* Serbia: Health System Review. *Health Syst Transit* **21**, 1-211 (2019).
- 96 MENG Qingyue, Y. H., CHEN Wen, SUN Qiang, LIU Xiaoyun. People's Republic of China health system review. *Health systems in transition* **5** (2015).
- 97 OECD. *OECD Reviews of Health Systems: Kazakhstan 2018*. (2018).
- 98 Nazer, L. H. & Tuffaha, H. Health Care and Pharmacy Practice in Jordan. *Can J Hosp Pharm* **70**, 150-155 (2017). <https://doi.org/10.4212/cjhp.v70i2.1649>
- 99 Fleifel, M. & Abi Farraj, K. The Lebanese Healthcare Crisis: An Infinite Calamity. *Cureus* **14**, e25367 (2022). <https://doi.org/10.7759/cureus.25367>
- 100 Tatar, M. *et al.* Turkey. Health system review. *Health Syst Transit* **13**, 1-186, xiii-xiv (2011).
- 101 Pongpisut Jongudomsuk, S. S., Walaiporn Patcharanarumol, Supon Limwattananon, Supasit Pannarunothai, Patama Vapatanavong, Krisada Sawaengdee, Pinij Fahamnuaypol. The Kingdom of Thailand health system review. *Health systems in transition* **5** (2015).

- 102 Government of Canada. *Algeria health sector market profile*, <<https://www.tradecommissioner.gc.ca/algeria-algerie/market-reports-etudes-de-marches/0006431.aspx?lang=eng>> (2022).
- 103 Zwarenstein, M. The structure of South Africa's health service. *Afr Health*, 3-4 (1994).
- 104 Dayrit, M. M., Lagrada, L.P., Picazo, O.F., Pons, M.C. & Villaverde, M.C. The Philippines Health System Review. *Health Systems in Transition* **8** (2018).
- 105 Selvaraj S, K. K. A., Srivastava S, Bhan N, & Mukhopadhyay I. India health system review. *Health systems in transition* **11** (2022).
- 106 Kurji, Z., Premani, Z. S. & Mithani, Y. Analysis Of The Health Care System Of Pakistan: Lessons Learnt And Way Forward. *J Ayub Med Coll Abbottabad* **28**, 601-604 (2016).
- 107 Fasseeh, A. *et al.* Healthcare financing in Egypt: a systematic literature review. *J Egypt Public Health Assoc* **97**, 1 (2022). <https://doi.org/10.1186/s42506-021-00089-8>
- 108 Brown, D. B., Chibi, M. T., Hassani, N., Smith, S. C. & Searles, R. V. Moroccan Health Care: A Link to Radicalization and Proposed Solution. *Fed Pract* **36**, 510-513 (2019).
- 109 Chahed, M. K. & Arfa, C. Monitoring and evaluating progress towards Universal Health Coverage in Tunisia. *PLoS Med* **11**, e1001729 (2014). <https://doi.org/10.1371/journal.pmed.1001729>
- 110 Government of Canada. *Prescription drug insurance coverage*, <<https://www.canada.ca/en/health-canada/services/health-care-system/pharmaceuticals/access-insurance-coverage-prescription-medicines.html>> (2020).
- 111 The Government of British Columbia. *BC PharmaCare: Get help paying for medications and medical supplies*, <<https://www2.gov.bc.ca/gov/content/health/health-drug-coverage/pharmacare-for-bc-residents>> (2023).
- 112 Government of Ontario. *Get coverage for prescription drugs*, <<https://www.health.gov.on.ca/en/public/programs/drugs/>> (2023).
- 113 Government of Ontario. *Check medication coverage*, <<https://www.ontario.ca/check-medication-coverage/>> (2022).
- 114 Katherine Keisler-Starkey, L. N. B. Health Insurance Coverage in the United States: 2021. Report No. P60-278, (United States, 2022).
- 115 U.S. Department of Health and Human Services. *Who's eligible for Medicare?*, <<https://www.hhs.gov/answers/medicare-and-medicaid/who-is-eligible-for-medicare/index.html>> (2022).
- 116 Medicare.gov. *Add prescription drug*, <https://www.medicare.gov/plan-compare/#/manage-prescriptions?fips=06037&plan_type=&year=2023&lang=en> (2023).
- 117 Medicaid.gov. *About Us*, <<https://www.medicaid.gov/about-us/index.html>> (2023).
- 118 Medicaid.gov. *Medicaid Drug Rebate Program (MDRP)*, <<https://www.medicaid.gov/medicaid/prescription-drugs/medicaid-drug-rebate-program/index.html#:~:text=The%20Medicaid%20Drug%20Rebate%20Program,drugs%20dispensed%20to%20Medicaid%20patients>> (2022).
- 119 Medicaid.gov. *Cost Sharing Out of Pocket Costs*, <<https://www.medicaid.gov/medicaid/cost-sharing/cost-sharing-out-pocket-costs/index.html>> (2023).
- 120 NYRx the New York Medicaid Pharmacy Program. *the Medicaid Pharmacy Program Preferred Drug List* <https://newyork.fhsc.com/downloads/providers/NYRx_PDP_PDL.pdf> (2023).
- 121 Mississippi Division of Medicaid. *Universal Preferred Drug List*, <<https://medicaid.ms.gov/preferred-drug-list/>> (2023).
- 122 World Health Organization. Regional Office for Europe. *Medicines reimbursement policies in Europe*. (World Health Organization. Regional Office for Europe., Denmark, 2018).
- 123 Health Service Executive (HSE), I. *Reimbursable Items*, <<https://www.ssps.ie/druglist/pub>> (2023).
- 124 Bundesinstitut für Arzneimittel und Medizinprodukte. *ABDA Festbetragsrecherche*, <<https://portal.dimdi.de/festbetragsrecherche/>> (2023).

- 125 North West London Integrated Care System. (ed United Kingdom National Health Service) (United Kingdom 2022).
- 126 NHS Business Services Authority United Kingdom. *NHS Prescription Prepayment Certificate (PPC)*, <<https://www.nhsbsa.nhs.uk/help-nhs-prescription-costs/nhs-prescription-prepayment-certificate-ppc>> (2023).
- 127 NHS inform Scotland. *Prescription charges and exemptions*, <<https://www.nhsinform.scot/care-support-and-rights/nhs-services/pharmacy/prescription-charges-and-exemptions>> (2023).
- 128 Welsh Government. *Free prescriptions*, <<https://www.gov.wales/free-prescriptions>> (2020).
- 129 Northern Ireland government. *Help with health costs*, <<https://www.nidirect.gov.uk/articles/help-health-costs#:~:text=All%20prescriptions%20dispensed%20in%20Northern,as%20everyone%20is%20automatically%20entitled.>> (2023).
- 130 Glover, L. *International Health Care System Profile - Australia*, <<https://www.commonwealthfund.org/international-health-policy-center/countries/australia#care-delivery-and-payment>> (2020).
- 131 Australian Government - Department of Health and Aged Care. *Pharmaceutical Benefits Scheme (PBS)*, <<https://www.pbs.gov.au/pbs/home>> (2023).
- 132 Ministry of Health New Zealand. *Prescription charges and the prescription subsidy scheme*, <<https://www.health.govt.nz/your-health/conditions-and-treatments/treatments-and-surgery/medications/prescription-charges-and-prescription-subsidy-scheme#:~:text=Many%20medicines%20in%20New%20Zealand,for%20prescriptions%20by%20approved%20providers>> (2023).
- 133 Pharmac New Zealand. *Community Schedule*, <<https://schedule.pharmac.govt.nz/ScheduleOnline.php>> (2023).
- 134 人力资源社会保障部, 国. 《国家基本医疗保险、工伤保险和生育保险药品目录(2022年)》的通知, <https://www.gov.cn/zhengce/zhengceku/2023-01/18/content_5737840.htm> (2023).
- 135 中华人民共和国人力资源和社会保障部. 中华人民共和国社会保险法释义(十二), <http://www.mohrss.gov.cn/fgs/syshehuibaixianfa/201208/t20120807_28573.html> (2012).
- 136 Fang, H. *International Health Care System Profile - China*, <<https://www.commonwealthfund.org/international-health-policy-center/countries/china>> (2020).
- 137 衛生福利部中央健康保險署. 2022-2023 全民健康保險年報. (2022).
- 138 Matsuda, R. *International Health Care System Profiles - Japan*, <<https://www.commonwealthfund.org/international-health-policy-center/countries/japan>> (2020).
- 139 厚生労働省. 中央社会保険医療協議会 総会(第545回)議事次第, <https://www.mhlw.go.jp/stf/shingi2/0000212500_00186.html> (2020).
- 140 Abe, Y. *Japan's NHI Drug Price System*, <<https://www.pmda.go.jp/files/000248690.pdf>> (2022).
- 141 Mahlich, J. & Sruamsiri, R. Co-insurance and health care utilization in Japanese patients with rheumatoid arthritis: a discontinuity regression approach. *Int J Equity Health* **18**, 22 (2019). <<https://doi.org/10.1186/s12939-019-0920-7>>
- 142 Health Insurance Review & Assessment Service. *Healthcare System in Korea*, <<https://www.hira.or.kr/dummy.do?pgmid=HIRAJ010000006000#:~:text=The%20health%20security%20system%20in,healthcare%20coverage%20to%20all%20citizens>> (
- 143 Park, J.-H. *Social security contributions*, <<https://taxsummaries.pwc.com/republic-of-korea/individual/other-taxes>> (2022).
- 144 Health Insurance Review & Assessment Service. *가 가 가 가 본인부담기준 안가*, (2022).

- 145 Health Insurance Review & Assessment Service. *가 가 가 가 가 가 가 가 가 가 (DUR)*,
<<https://www.hira.or.kr/ra/medi/form.do?pgmid=HIRAA030029000000&WT.gnb=%EC%9D%98%EC%95%BD%ED%92%88%EC%95%88%EC%A0%84%EC%82%AC%EC%9A%A9%EC%84%9C%EB%B9%84%EC%8A%A4%28DUR%29>> (
- 146 National Institute for Health and Care Excellence. Generalised anxiety disorder and panic disorder in adults: management. (2022).
- 147 National Institute for Health and Care Excellence. Epilepsies in children, young people and adults. (2022).
- 148 National Institute for Health and Care Excellence. Neuropathic pain in adults: pharmacological management in non-specialist settings. (2013).
- 149 Baldwin, D. S. *et al.* Evidence-based pharmacological treatment of anxiety disorders, post-traumatic stress disorder and obsessive-compulsive disorder: a revision of the 2005 guidelines from the British Association for Psychopharmacology. *J Psychopharmacol* **28**, 403-439 (2014). <https://doi.org:10.1177/0269881114525674>
- 150 National Institute for Health and Care Excellence. Restless legs syndrome: Oxycodone/naloxone prolonged release. (2015).
- 151 Attal, N. *et al.* EFNS guidelines on the pharmacological treatment of neuropathic pain: 2010 revision. *Eur J Neurol* **17**, 1113-e1188 (2010). <https://doi.org:10.1111/j.1468-1331.2010.02999.x>
- 152 Katzman, M. A. *et al.* Canadian clinical practice guidelines for the management of anxiety, posttraumatic stress and obsessive-compulsive disorders. *BMC Psychiatry* **14 Suppl 1**, S1 (2014). <https://doi.org:10.1186/1471-244X-14-S1-S1>
- 153 Moulin, D. *et al.* Pharmacological management of chronic neuropathic pain: revised consensus statement from the Canadian Pain Society. *Pain Res Manag* **19**, 328-335 (2014). <https://doi.org:10.1155/2014/754693>
- 154 Laura J. Fochtmann. PRACTICE GUIDELINE FOR THE Treatment of Patients With Panic Disorder. (American Psychiatric Association (APA), 2009).
- 155 Bril, V. *et al.* Evidence-based guideline: Treatment of painful diabetic neuropathy: report of the American Academy of Neurology, the American Association of Neuromuscular and Electrodiagnostic Medicine, and the American Academy of Physical Medicine and Rehabilitation. *Neurology* **76**, 1758-1765 (2011). <https://doi.org:10.1212/WNL.0b013e3182166ebe>
- 156 Chou, R. *et al.* Management of Postoperative Pain: A Clinical Practice Guideline From the American Pain Society, the American Society of Regional Anesthesia and Pain Medicine, and the American Society of Anesthesiologists' Committee on Regional Anesthesia, Executive Committee, and Administrative Council. *J Pain* **17**, 131-157 (2016). <https://doi.org:10.1016/j.jpain.2015.12.008>
- 157 Kanner, A. M. *et al.* Practice guideline update summary: Efficacy and tolerability of the new antiepileptic drugs I: Treatment of new-onset epilepsy: Report of the American Epilepsy Society and the Guideline Development, Dissemination, and Implementation Subcommittee of the American Academy of Neurology. *Epilepsy Curr* **18**, 260-268 (2018). <https://doi.org:10.5698/1535-7597.18.4.260>
- 158 Kanner, A. M. *et al.* Practice guideline update summary: Efficacy and tolerability of the new antiepileptic drugs II: Treatment-resistant epilepsy: Report of the American Epilepsy Society and the Guideline Development, Dissemination, and Implementation Subcommittee of the American Academy of Neurology. *Epilepsy Curr* **18**, 269-278 (2018). <https://doi.org:10.5698/1535-7597.18.4.269>
- 159 Bandelow, B. *et al.* Guidelines for the pharmacological treatment of anxiety disorders, obsessive-compulsive disorder and posttraumatic stress disorder in primary care. *Int J Psychiatry Clin Pract* **16**, 77-84 (2012). <https://doi.org:10.3109/13651501.2012.667114>

160 Finnerup, N. B. *et al.* Pharmacotherapy for neuropathic pain in adults: a systematic review and meta-analysis. *Lancet Neurol* **14**, 162-173 (2015). [https://doi.org/10.1016/S1474-4422\(14\)70251-0](https://doi.org/10.1016/S1474-4422(14)70251-0)